# Learning Feature Sparse Principal Subspace

**Lai Tian**
School of Computer Science &
Center for OPTIMAL,
Northwestern Polytechnical University,
Xi'an 710072, China.
tianlai.cs@gmail.com

**Feiping Nie**[*]
School of Computer Science &
Center for OPTIMAL,
Northwestern Polytechnical University,
Xi'an 710072, China.
feipingnie@gmail.com

**Rong Wang**
School of Cybersecurity &
Center for OPTIMAL,
Northwestern Polytechnical University,
Xi'an 710072, China.
wangrong07@tsinghua.org.cn

**Xuelong Li**
School of Computer Science &
Center for OPTIMAL,
Northwestern Polytechnical University,
Xi'an 710072, China.
li@nwpu.edu.cn

## Abstract

This paper presents new algorithms to solve the feature-sparsity constrained PCA problem (FSPCA), which performs feature selection and PCA simultaneously. Existing optimization methods for FSPCA require data distribution assumptions and lack of global convergence guarantee. Though the general FSPCA problem is NP-hard, we show that, for a low-rank covariance, FSPCA can be solved globally (Algorithm 1). Then, we propose another strategy (Algorithm 2) to solve FSPCA for the general covariance by iteratively building a carefully designed proxy. We prove (data-dependent) approximation bound and convergence guarantees for the new algorithms. For the spectrum of covariance with exponential/Zipf's distribution, we provide exponential/posynomial approximation bound. Experimental results show the promising performance and efficiency of the new algorithms compared with the state-of-the-arts on both synthetic and real-world datasets.

## 1 Introduction

Consider $n$ data points in $\mathbb{R}^d$. When $d \gg n$, PCA has inconsistence issue in estimating the $m$ leading eigenvectors $\mathbf{W} \in \mathbb{R}^{d \times m}$ of population covariance matrix $\mathbf{A} \in \mathbb{R}^{d \times d}$ [18], which can be addressed by assuming the sparsity in the principal components. Prior work has been done in methodology design [51, 38, 11, 41, 36, 47, 34, 33, 22] and theoretical understanding [42, 23, 46, 50].

The principal subspace estimation [6, 20, 28, 16, 45] is directly connected to dimension reduction and is important when there are more than one principal component of interest. Indeed, typical applications of PCA use the projection onto the principal subspace to facilitate exploration and inference of important features of the data. As Vu et al. [42] point out, dimension reduction by PCA should emphasize subspaces rather than eigenvectors. The sparsity level in sparse principal subspace estimation is defined as follows [42, 43, 47].

**Definition 1.1** (Subspace sparsity, [42]). *For the $m$-dimensional principal subspace* span($\mathbf{W}$) *of the covariance* $\mathbf{A}$*, the subspace sparsity level $k$ is defined by*

$$k = \mathrm{card}(\mathrm{supp}[\mathrm{diag}(\boldsymbol{\Pi})]) = \|\mathbf{W}\|_{2,0},$$

---

[*]Corresponding author: Feiping Nie

$$\mathbf{W}_{(a)}^\top = \begin{bmatrix} \boxed{-0.3} & 0.0 & \boxed{-0.7} & \boxed{-0.7} & 0.0 \\ 0.0 & \boxed{-0.8} & 0.0 & \boxed{-0.2} & \boxed{0.5} \\ 0.0 & \boxed{-0.5} & \boxed{0.7} & 0.0 & \boxed{-0.5} \end{bmatrix} \begin{matrix} 1 \\ \vdots \\ m \end{matrix} \qquad \mathbf{W}_{(b)}^\top = \begin{bmatrix} \boxed{-0.6} & 0.0 & \boxed{-0.6} & 0.0 & \boxed{-0.5} \\ \boxed{-0.6} & 0.0 & \boxed{0.8} & 0.0 & \boxed{-0.1} \\ \boxed{0.4} & 0.0 & \boxed{0.3} & 0.0 & \boxed{-0.9} \end{bmatrix} \begin{matrix} 1 \\ \vdots \\ m \end{matrix}$$

Figure 1: Element-wise Sparse PCA $\mathbf{W}_{(a)}$ versus Feature Sparse PCA $\mathbf{W}_{(b)}$.

*where* $\mathbf{\Pi} = \mathbf{W}\mathbf{W}^\top$ *is the projection matrix onto* $\mathrm{span}(\mathbf{W})$ *and* $\|\cdot\|_{2,0}$ *is the row-sparsity norm.*

This paper considers the principal subspace estimation problem with the feature subspace sparsity constraint, termed Feature Sparse PCA (Problem (3.1) ). Some approaches have been proposed to solve the FSPCA problem [43, 47, 27]. Yet, there are some drawbacks in the existing methods. (1) Most of the existing analysis only holds in high probability when specific data generation assumptions hold, e.g., Yang & Xu [47] requires data generated from the spike model, Wang et al. [43] requires data generated from the sub-Gaussian distribution. Otherwise, they only guarantee convergence when the initial solution is near the global optimum. (2) In practice, monotonic algorithms are preferred as they bring improvement in every step. However, existing iterative schemes for FSPCA are not ascent guaranteed. (3) Some methods make the spike model assumption, in which the population covariance is instinctively low-rank (up to an additive scaled identity), but existing methods cannot make full use of the low-rank structure in the covariance.

Compared with prior work which mostly averaging out the worst case by assuming probability model on the covariance, our work provides algorithms with deterministic analysis from the optimization aspect which is in a model-free style, thus, can be applied to any model. [21, 9, 36, 1] also consider the sparse PCA problem from the optimization perspective. But they only compute the leading sparse eigenvector, which might be suboptimal when multiple eigenvectors are considered.

In this paper, we provide two optimization strategies to compute the leading sparse principal subspace with provable optimization guarantees. The first one (Algorithm 1) solves the feature sparse PCA problem globally when the covariance matrix is low-rank, while the second one (Algorithm 2) solves the feature sparse PCA for general covariance matrix iteratively with guaranteed convergence.

**Contributions.** More precisely, we make the following contributions:

1. We show that, for a low-rank covariance matrix, the FSPCA problem can be solved globally with the newly proposed algorithm (Algorithm 1). For the general high-rank case, we report an iterative algorithm (Algorithm 2) by building a carefully designed proxy.

2. We prove (data-dependent) approximation bound and convergence guarantees for the proposed optimization strategies. Computational complexities of both algorithms are analyzed.

3. We conduct experiments on both synthetic and real-world data to evaluate the new algorithms. The experimental results demonstrate the promising performance of the newly proposed algorithms compared with the state-of-the-art methods.

**Notations.** Throughout this paper, scalars, vectors and matrices are denoted by lowercase letters, boldface lowercase letters and boldface uppercase letters, respectively; for a matrix $\mathbf{A} \in \mathbb{R}^{d \times d}$, $\mathbf{A}^\top$ denotes the transpose of $\mathbf{A}$, $\mathrm{Tr}(\mathbf{A}) = \sum_{i=1}^{d} a_{ii}$, $\|\mathbf{A}\|_F^2 = \mathrm{Tr}(\mathbf{A}^\top \mathbf{A})$; $\mathbb{1}_{\{\text{condition}\}}$ is the $(0,1)$-indicator of the condition; $\mathbb{1}_n \in \mathbb{R}^n$ denotes vector with all ones; $\|\mathbf{x}\|_0$ denotes the number of non-zero elements; $\|\mathbf{W}\|_{2,0} = \sum_{i=1}^{d} \|\mathbf{w}_i\|_2^0 = \sum_{i=1}^{d} \mathbb{1}\{\|\mathbf{w}_i\| \neq 0\}$ measures the row-sparsity of $\mathbf{W}$ where $\mathbf{W} \in \mathbb{R}^{d \times m}$, $\mathbf{w}_i \in \mathbb{R}^{1 \times m}$ is the $i$th row of $\mathbf{W}$; $\mathbb{I}_{n \times n} \in \mathbb{R}^{n \times n}$ denotes the identity matrix; $\mathcal{I}(1:k)$ is the first $k$ elements in indices $\mathcal{I}$; $\mathbf{A}^\dagger$ denotes the Moore–Penrose inverse; $\mathbf{A}_m$ is the best rank-$m$ approximation of $\mathbf{A}$ in Frobenius norm; $\mathrm{card}(\mathcal{I})$ is the cardinality of $\mathcal{I}$; $[n] := \mathbb{Z} \cap \{i : 1 \leq i \leq n\}$. We assume that the eigenvalues $\{\lambda_i\}_{i=1}^n$ are arranged in descending order, i.e., $\lambda_1 \geq \lambda_2 \geq \cdots \geq \lambda_n$.

## 2 Prior Work

In this section, we review several prior arts that consider related problems.

**Sparse Principal Components.** Most existing methods in the literature to solve the sparse PCA problem only estimate the first leading eigenvector with the element-wise sparsity constraint. To estimate the $m$ leading eigenvectors, one has to build a new covariance matrix with the deflation technique [26] and solve the leading eigenvector again. The main drawback of this scheme is that, for example, the indices of non-zero elements in the first eigenvector might not be the same as that of the second eigenvector. As shown in Figure 1, the sparsity pattern is inconsistent among the $m$ leading eigenvectors, which causes difficulties in applications, e.g., feature selection. Moreover, the deflation has identifiability and orthogonality issues when the top $m$ eigenvalues are not distinct [43]. [1, 36, 21, 9] propose methods and analysis for the leading eigenvector with approximation guarantee but their guarantee only applies to the first component, not to further deflation iterations.

**Sparse Principal Subspace.** Vu et al. [42] consider a different setting that the estimated subspace is subspace sparsity constrained (Definition 1.1), in which the sparsity pattern is forced consistent among rows. They show this problem has nice statistical properties [42], that is, the optimum is statistically minimax optimal. But there is a gap between the computational method and statistical theory. To close this gap, [43, 47, 27, 6, 20, 28, 16, 45] proposed algorithms to solve the subspace sparsity constrained problem. However, from an optimization viewpoint, existing methods require data distribution assumptions and lack of global convergence guarantee. Besides, [2] proposed an algorithm that runs exponential in the rank($\mathbf{A}$) and $m$ for the *disjoint*-FSPCA problem that requires the support of different eigenvectors to be disjoint, which is clearly different from our setting.

**Sparse Regression.** Another line of research [35, 13, 8, 32] considers solving the sparse regression problem with the $\ell_{2,0}$ constraint or its convex relaxation. The main technical difference between the $\ell_{2,0}$ constrained sparse regression and FSPCA is the semi-orthogonal constraint on $\mathbf{W}$. Without the semi-orthogonal constraint, the FSPCA problem is not bound from above. Existing techniques to solve the $\ell_{2,0}$ constrained sparse regression problem, e.g., the projected gradient scheme in [35], cannot be used to solve our problem because, to our knowledge, there is no method to solve the projection subproblem with the semi-orthogonal constraint. Thus, the FSPCA problem is substantially more difficult than that of $\ell_{2,0}$-constrained sparse regression.

## 3 Problem Setup

Formally, we propose algorithms to solve the following general problem

$$\max_{\mathbf{W} \in \mathbb{R}^{d \times m}} \mathrm{Tr}\left(\mathbf{W}^\top \mathbf{A} \mathbf{W}\right) \ \text{ s.t. } \ \mathbf{W}^\top \mathbf{W} = \mathbb{I}_{m \times m}, \|\mathbf{W}\|_{2,0} \leq k, \tag{3.1}$$

where $m \leq k \leq d$ and matrix $\mathbf{A} \in \mathbb{R}^{d \times d}$ is positive semi-definite. This problem is NP-hard to solve globally even for $m = 1$ [30] and sadly NP-hard to solve $(1 - \varepsilon)$-approximately for a small $\mu > 0$ if $\varepsilon < \mu$ [9]. Several techniques have been proposed [43, 47] to solve this challenging problem. However, they only report high-probability analysis and none of them provides practical algorithm with deterministic guarantee on both approximation and global convergence.

**Remark 3.1.** *As shown in Vu et al. [42], the optimal $\mathbf{W}$ of Problem (3.1) achieves the optimal minimax error for row sparse subspace estimation. Besides, the FSPCA problem can be viewed as performing unsupervised feature selection and PCA simultaneously. The key point is the $\ell_{2,0}$ norm constraint forces the sparsity pattern consistence among different eigenvectors, while the vanilla element-wise sparse PCA model cannot keep this consistence as shown in Figure 1. One might use only the leading sparse eigenvector for feature selection [25, 31] but this leads to suboptimal solution when there are more than one principal component of interest (see Figure 2, TPower (G) ).*

## 4 Optimization Strategies

In this section, we provide new optimization strategies to solve the FSPCA model in Problem (3.1). We first consider the case when rank($\mathbf{A}$) $\leq m$, for which a non-iterative strategy (Algorithm 1) is provided to solve the problem globally. Then we consider the general case when rank($\mathbf{A}$) $> m$, for which we provide an iterative algorithm (Algorithm 2) by approximating $\mathbf{A}$ with a carefully designed low-rank proxy covariance $\mathbf{P}$ and solve the proxy subproblem with the Algorithm 1.

## 4.1 GO: Global Optimum if $\text{rank}(\mathbf{A}) \leq m$

We make the following notion for ease of notations.

**Definition 4.1** (Row selection matrix map). *We use $(d, k)$-row selection matrix map $\mathbb{S}_{d,k}(\mathcal{I})$ to build row selection matrix $\mathbf{S} \in \mathbb{R}^{d \times k}$ according to given indices $\mathcal{I}$ such that $\mathbb{S}_{d,k}(\mathcal{I}) = \mathbf{S}$, i.e., $s_{ij} = \mathbb{1}_{i=\mathcal{I}(j)}$. One can left multiply the selection matrix $\mathbf{S}$ to select specific $k$ rows from $d$ inputs.*

The algorithm to solve Problem (3.1) is summarized in the following Algorithm 1.

---

**Algorithm 1** Go for $\text{rank}(\mathbf{A}) \leq m$

---

1: **procedure** GO$(\mathbf{A}, m, k, d)$
2:     $\mathcal{I} \leftarrow$ indices of the $k$ largest elements of diag$(\mathbf{A})$        ▷ prefer smaller indices if tied.
3:     $\mathbf{S} \leftarrow \mathbb{S}_{d,k}(\mathcal{I})$;
4:     $\mathbf{V} \leftarrow m$ first eigenvectors of $\mathbf{A}_{\mathcal{I},\mathcal{I}}$
5:     **return** $\mathbf{W} \leftarrow \mathbf{SV}$;
6: **end procedure**

---

The following theorem justifies the **global optimality** of the output of Algorithm 1:

**Theorem 4.2.** *Suppose $\mathbf{A} \succcurlyeq \mathbb{0}$ and $\text{rank}(\mathbf{A}) \leq m$. Let $\mathbf{W} = \text{GO}(\mathbf{A}, m, k, d)$ with $m \leq k \leq d$. Then, $\mathbf{W}$ is a globally optimal solution of Problem (3.1).*

**Remark 4.3.** *Theorem 4.2 guarantees the global optimality of Algorithm 1 for a low-rank $\mathbf{A}$. It is interesting to see that, though the Problem (3.1) is NP-hard to solve in general, it is globally solvable for a low-rank covariance $\mathbf{A}$. A natural idea then comes out that we can try to solve the general Problem (3.1) by running Algorithm 1 with the best rank-$m$ approximation $\mathbf{A}_m$. In Theorem 5.1 and Section 6, we will justify this idea theoretically and empirically.*

**Remark 4.4.** *It is notable that, for any $\mathbf{B} \in \{\mathbf{A} + \sigma \mathbb{I}_{d \times d} : \mathbf{A} \succcurlyeq \mathbb{0}, \text{rank}(\mathbf{A}) \leq m, \sigma \geq 0\}$, which is the population covariance in the spike model, the Algorithm 1 still outputs a globally optimal solution with $\mathbf{B} - \sigma \mathbb{I}_{d \times d}$ as the input, since $\text{Tr}\left(\mathbf{W}^\top \mathbf{B} \mathbf{W}\right) = \text{Tr}\left(\mathbf{W}^\top (\mathbf{B} - \sigma \mathbb{I}_{d \times d}) \mathbf{W}\right) + \sigma m = \text{Tr}\left(\mathbf{W}^\top \mathbf{A} \mathbf{W}\right) + \sigma m$. Sufficient condition $\text{rank}(\mathbf{A}) \leq m$ is a special case that $\sigma = 0$.*

## 4.2 IPU: Iteratively Proxy Update for $\text{rank}(\mathbf{A}) > m$

In this subsection, we consider the general case, that is, $\text{rank}(\mathbf{A}) > m$. The main idea is that we try to build a proxy covariance, say $\mathbf{P}$, of original $\mathbf{A}$ such that $\text{rank}(\mathbf{P}) \leq m, \mathbf{P} \succcurlyeq \mathbb{0}$. Then we can run Algorithm 1 with the low-rank proxy $\mathbf{P}$ to solve the original problem iteratively. Besides, we note here the proxy covariance $\mathbf{P}$ introduced below, by design, makes the iterative procedure an MM-type one, which directly suggests its convergence by construction (see Section 5.2 for details).

**Proxy Construction.** With careful design, given the estimate $\mathbf{W}_t$ from the $t$th iterative step, we define the matrix

$$\mathbf{P}_t = \mathbf{A} \mathbf{W}_t (\mathbf{W}_t^\top \mathbf{A} \mathbf{W}_t)^\dagger \mathbf{W}_t^\top \mathbf{A}$$

as the low-rank proxy matrix of original $\mathbf{A}$. Then, we solve Problem 3.1 with the proxy $\mathbf{P}_t$ rather than $\mathbf{A}$. Following claim verifies the sufficient conditions for $\mathbf{P}_t$ to be solvable with Algorithm 1.

**Claim 4.5.** *For each $t \geq 1$, $\mathbf{W}_t^\top \mathbf{W}_t = \mathbb{I}_{m \times m}$, it holds $\text{rank}(\mathbf{P}_t) \leq m$, and $\mathbf{P}_t \succcurlyeq \mathbb{0}$.*

**Indices Selection.** With the proxy matrix $\mathbf{P}_t$ in hand, a natural idea is to iteratively update $\mathbf{W}$ by solving the following problem with Algorithm 1:

$$\widetilde{\mathbf{W}}_{t+1} \leftarrow \text{GO}(\mathbf{P}_t, m, k, d). \tag{4.1}$$

But we can further refine the $\widetilde{\mathbf{W}}_{t+1}$ by performing eigenvalue decomposition on original $\mathbf{A}$ rather than on the proxy covariance $\mathbf{P}_t$, which will accelerate the convergence.

**Eigenvectors Refinement.** Note that $\widetilde{\mathbf{W}}_{t+1}$ can be written as $\widetilde{\mathbf{W}}_{t+1} = \mathbf{S}_{t+1} \widetilde{\mathbf{V}}_{t+1}$, where $\mathbf{S}_{t+1}$ is the selection matrix and $\widetilde{\mathbf{V}}_{t+1}$ is the eigenvectors in the row support of $\widetilde{\mathbf{W}}_{t+1}$. Then, $\widetilde{\mathbf{W}}_{t+1}$ can be

further refined by fixing the selection matrix $\mathbf{S}_{t+1}$ and updating the eigenvectors $\mathbf{V}_{t+1}$ with

$$\mathbf{V}_{t+1} \leftarrow \underset{\mathbf{V}^\top \mathbf{V} = \mathbb{I}_{m \times m}}{\arg \max} \operatorname{Tr}(\mathbf{V}^\top \mathbf{S}_{t+1}^\top \mathbf{A} \mathbf{S}_{t+1} \mathbf{V}). \tag{4.2}$$

Finally, the refined $\mathbf{W}_{t+1}$ can be computed by $\mathbf{W}_{t+1} \leftarrow \mathbf{S}_{t+1} \mathbf{V}_{t+1}$. Compared with updating with Problem (4.1), updating with the refinement makes larger progress thus it is more aggressive.

---

**Algorithm 2** IPU for general $\mathbf{A}$

---

1: **procedure** IPU($\mathbf{A}, m, k, d, \mathbf{W}_0$)
2:     $t \leftarrow 0$;
3:     **repeat**
4:         $\mathbf{P}_t \leftarrow \mathbf{A} \mathbf{W}_t (\mathbf{W}_t^\top \mathbf{A} \mathbf{W}_t)^\dagger \mathbf{W}_t^\top \mathbf{A}$;
5:         $[\mathbf{S}, \mathcal{I}] \leftarrow \texttt{Go}(\mathbf{P}_t, m, k, d)$;
6:         $\mathbf{V} \leftarrow m$ first eigenvectors of $\mathbf{A}_{\mathcal{I},\mathcal{I}}$
7:         $\mathbf{W}_{t+1} \leftarrow \mathbf{S}\mathbf{V}$;    $t \leftarrow t + 1$;
8:     **until** $\mathbf{W}_t = \mathbf{W}_{t-1}$
9:     **return** $\mathbf{W}_t$;
10: **end procedure**

---

In summary, we collect the procedure to solve FSPCA when $\operatorname{rank}(\mathbf{A}) > m$ in Algorithm 2. The iterative procedure in Algorithm 2 is simple and well-motivated by the iteratively updated proxy idea. However, existing algorithms [43, 47] in the literature usually follow the orthogonal iteration scheme [19], which makes it hard to see the difference between prior arts and IPU. To cope with this, we provide an orthogonal iteration like reformulation of Algorithm 2 and a detailed discussion in Appendix due to space limitation, which might be of interest on its own.

## 5 Theoretical Analysis

In this section, we provide the theoretical analysis for Algorithm 1 and 2. In detail, we prove approximation and convergence guarantees for the new algorithms. Then, we report the computational complexities and compare them with these of methods in the literature.

### 5.1 Approximation Guarantee

The intuition guiding us to the approximation ratio bound is that, while we have global optimality if $\operatorname{rank}(\mathbf{A}) \leq m$, we want to understand the solution accuracy if we have the $\operatorname{rank}(\mathbf{A})$ "almost" $\leq m$.

To begin, we define constants related to the eigenvalues decay of $\mathbf{A}$. Let $r = \min\{\operatorname{rank}(\mathbf{A}), 2m\}$,

$$G_1 = \frac{\sum_{i=m+1}^{r} \lambda_i(\mathbf{A})}{\sum_{i=1}^{m} \lambda_i(\mathbf{A})}, \qquad G_2 = \frac{\sum_{i=m+1}^{r} \lambda_i(\mathbf{A})}{\sum_{i=1}^{d} \lambda_i(\mathbf{A})}.$$

The main approximation result can be stated as follows.

**Theorem 5.1.** *Suppose $\mathbf{A} \succcurlyeq \mathbb{0}$ with condition number $\kappa$, $m \leq k \leq d$. Let $\mathbf{W}_m = Go(\mathbf{A}_m, m, k, d)$, and $\mathbf{W}_*$ be globally optimal for Problem 3.1. Then, we have $(1 - \varepsilon) \leq \frac{\operatorname{Tr}(\mathbf{W}_m^\top \mathbf{A} \mathbf{W}_m)}{\operatorname{Tr}(\mathbf{W}_*^\top \mathbf{A} \mathbf{W}_*)} \leq 1$ with*

$$\varepsilon \leq \min\left\{\frac{dG_1}{k}, \frac{dG_2}{m}, 1 - \kappa^{-1}, 1 - \frac{k}{d}\right\}.$$

**Remark 5.2.** *Theorem 5.1 says that, for sufficiently large $m$ or $k$, $Go(\mathbf{A}_m, m, k, d)$ gives a certified approximate solution of Problem 3.1. Also note that, when the eigenvalues of the covariance $\mathbf{A}$ decay fast enough, a small $m$ or $k$ is sufficient to guarantee certified approximation. It is notable that, when $\operatorname{rank}(\mathbf{A}) \leq m$, we have $G_1 = G_2 = 0$, $\mathbf{A} = \mathbf{A}_m$, which implies $\varepsilon = 0$ and the output of the Algorithm 1 is globally optimal. Using Theorem 4.2, the bound given in Theorem 5.1 is sharp.*

If the eigenvalues of $\mathbf{A}$ decay sufficiently fast, e.g., exponentially, the bound would be tighter.

**Corollary 5.3** (Exponential distribution). *Suppose $\mathbf{A} \succcurlyeq \mathbb{0}, m \leq k \leq d$, and $\lambda_i(\mathbf{A}) = c'e^{-ci}$ with $c' > 0, c > 0$ for each $i = 1, \ldots, 2m$. Let $\mathbf{W}_m = Go(\mathbf{A}_m, m, k, d)$, and $\mathbf{W}_*$ be an optimal solution of Problem 3.1. If $m \geq \Omega\left(\frac{1}{c} \log\left(\frac{d}{k\varepsilon}\right)\right)$, then we have $(1 - \varepsilon) \leq \frac{\operatorname{Tr}(\mathbf{W}_m^\top \mathbf{A} \mathbf{W}_m)}{\operatorname{Tr}(\mathbf{W}_*^\top \mathbf{A} \mathbf{W}_*)} \leq 1$.*

The difficult case is when the spectrum of $\mathbf{A}$ has a heavy-tail distribution, e.g., Zipf's law, a.k.a., Pareto's distribution. It has been observed by Breslau et al. [7], Faloutsos et al. [14], Mihail & Papadimitriou [29] that many phenomena approximately follow Zipf-like spectrum, e.g., Web caching, Internet topology, and city population. The $i$th eigenvalue of the Zipf-like spectrum is $ci^{-t}$ with constants $c > 0, t > 1$. We have following corollary for Zipf-like distributed eigenvalues.

**Corollary 5.4** (Zipf's distribution). *Suppose $\mathbf{A} \succcurlyeq \mathbb{0}, m \leq k \leq d$, and $\lambda_i(\mathbf{A}) = ci^{-t}$ with $t > 1, c > 0$ for each $i = 1, \ldots, 2m$. Let $\mathbf{W}_m = Go(\mathbf{A}_m, m, k, d)$, and $\mathbf{W}_*$ be an optimal solution of Problem 3.1. If $m \geq \Omega\left(\left(\frac{d}{k\varepsilon}\right)^{\frac{1}{t-1}}\right)$, then we have $(1 - \varepsilon) \leq \frac{\operatorname{Tr}(\mathbf{W}_m^\top \mathbf{A} \mathbf{W}_m)}{\operatorname{Tr}(\mathbf{W}_*^\top \mathbf{A} \mathbf{W}_*)} \leq 1$.*

## 5.2 Convergence Guarantee

In this section, we show the iterative scheme proposed in Algorithm 2 increases the objective function value in every iterative step, which directly indicates the convergence of the iterative scheme.

Lots of classical algorithms can be framed into the MM framework, e.g., EM Algorithm [12], Proximal Algorithms [4, 37], Concave-Convex Procedure (CCCP) [49, 24]. Please refer to [39] for further discussion. It is notable that the newly proposed Algorithm 2 can also be viewed as a special case of the general MM optimization framework. Unlike conventional MM using Jensen's/A-G-M/Cauchy-Schwartz's inequalities, or quadratic upper bound to build auxiliary function [44, 39], our auxiliary function for Algorithm 2 is based on the von Neumann's trace inequality [40], which is defined by $g(\mathbf{W}; \mathbf{W}_t) = \mathrm{Tr}(\mathbf{W}^\top \mathbf{A} \mathbf{W}_t (\mathbf{W}_t^\top \mathbf{A} \mathbf{W}_t)^\dagger \mathbf{W}_t^\top \mathbf{A} \mathbf{W}) \leq \mathrm{Tr}(\mathbf{W}^\top \mathbf{A} \mathbf{W})$. Meanwhile, it is easy to check that $g(\mathbf{W}; \mathbf{W}_t)$ satisfies $g(\mathbf{W}_t; \mathbf{W}_t) = \mathrm{Tr}(\mathbf{W}_t^\top \mathbf{A} \mathbf{W}_t)$.

**Theorem 5.5** (Monotonic increasing). *Suppose $\mathbf{A} \succcurlyeq 0, m \leq k \leq d$. Let $\mathbf{W}_{t+1}$ be the variable defined in Algorithm 2. If $\mathbf{W}_t \neq \mathbf{W}_{t+1}$ up to EVD, then, $\mathrm{Tr}(\mathbf{W}_t^\top \mathbf{A} \mathbf{W}_t) < \mathrm{Tr}(\mathbf{W}_{t+1}^\top \mathbf{A} \mathbf{W}_{t+1})$.*

Leveraging the ascent property, we have following the approximation guarantee for Algorithm 2.

**Corollary 5.6.** *Suppose $\mathbf{A} \succcurlyeq 0, \kappa = \lambda_1(\mathbf{A})/\lambda_d(\mathbf{A})$. Let $\widehat{\mathbf{W}} = IPU(\mathbf{A}, m, k, d, Go(\mathbf{A}_m, m, k, d))$, and $\mathbf{W}_*$ be an optimal solution of Problem 3.1. Then, we have $(1 - \varepsilon) \leq \frac{\mathrm{Tr}(\widehat{\mathbf{W}}^\top \mathbf{A} \widehat{\mathbf{W}})}{\mathrm{Tr}(\mathbf{W}_*^\top \mathbf{A} \mathbf{W}_*)} \leq 1$ with*

$$\varepsilon \leq \min\left\{\frac{dG_1}{k}, \frac{dG_2}{m}, 1 - \kappa^{-1}, 1 - \frac{k}{d}\right\}.$$

**Remark 5.7.** *Theorem 5.5 shows that the newly proposed Algorithm 2 is an ascent method, that is $\{\mathrm{Tr}(\mathbf{W}_t^\top \mathbf{A} \mathbf{W}_t)\}_{t=1}^T$ is an increasing sequence, which is important since most of the existing algorithms for solving Problem (3.1) are not ascent. That is to say, they cannot guarantee the output is better than the initialization. Combining with the fact that the objective function is bounded from above by finite $\mathrm{Tr}(\mathbf{A})$, the convergence of objective function value can be obtained.*

We show the sequence from Algorithm 2 converges to a fixed point in the sense of subspace.

**Theorem 5.8** (Convergence). *Suppose $\mathbf{A} \succcurlyeq 0, m \leq k \leq d$, and $\lambda_m - \lambda_{m+1} > 0$ on the selected principal submatrix of fixed point. Let $\{\mathbf{W}_t\}_{t=1}^\infty$ be any sequence generated by Algorithm 2. Then, the sequence $\{\mathbf{W}_t\}_{t=1}^\infty$ converges to a fixed point, say $\widetilde{\mathbf{W}}$, of Algorithm 2 in the sense of subspace, and $\|\sin\Theta(\mathrm{span}(\mathbf{W}_{t+1}), \mathrm{span}(\mathbf{W}_t))\|_2 \to 0, \mathrm{Tr}(\mathbf{W}_t^\top \mathbf{A} \mathbf{W}_t) \to \mathrm{Tr}(\widetilde{\mathbf{W}}^\top \mathbf{A} \widetilde{\mathbf{W}})$.*

## 5.3 Computational Complexity

**Algorithm 1.** It is easy to see the overall complexity is $O(d + k^3)$ since $O(d)$ for the largest $k$ indices selection (use the $\Theta(d)$ median of medians [5] to select the largest $k$-th element, then do a scan to filter elements that is larger than the $k$-th element), $O(k^3)$ for eigenvalue decomposition, and $O(km)$ for building the output $\mathbf{W}$.

**Algorithm 2.** The overall computational complexity[2] is $O(\max\{dkm, k^3\}T)$, where $T$ is the number of iterative steps used to coverage. We did not provide an upper bound on $T$ as characterizing the rate of convergence for most MM algorithm is very hard [44] (except for some quadratic upper bound type algorithms). But we empirically observe in Section 6.1 that $T \leq 10$ for both synthetic and real-world data. For proxy covariance construction and indices selection, we need $O(d^2m)$ for naively building $\mathbf{P}_t$ and running Algorithm 1. But note that we only need the diagonal elements in $\mathbf{P}_t$ for sorting and selecting. Thus, we only compute the diagonal elements of $\mathbf{P}_t$ and sort it for the indices selection[3], that is $O(dkm)$. Then, performing eigenvectors refinement and updating $\mathbf{W}_{t+1}$ costs $O(k^3)$. Also note that, the computational complexity of SOAP proposed in [43] is $O(d^2m)$ for every iterative step. Ours computational complexity is strictly less than that of SOAP. For SRT in [47], the computational complexity is $O(dm\min\{m, k\log d\})$. When $k = O(m)$, our complexity matches that of SRT.

Table 1: Synthetic Data Description

| No. | Description | Note |
|---|---|---|
| A | $\lambda(\mathbf{A}) = \{100, 100, 4, 1, \dots, 1\}$ | Setting in [43] |
| B | $\lambda(\mathbf{A}) = \{300, 180, 60, 1, \dots, 1\}$ | Setting in [43] |
| C | $\lambda(\mathbf{A}) = \{300, 180, 60, 0, \dots, 0\}$ | Verify the correctness of Theorem 4.2 |
| D | $\lambda(\mathbf{A}) = \{160, 80, 40, 20, 10, 5, 2, 1, \dots, 1\}$ | For all $\sigma$, $\mathrm{rank}(\mathbf{A} + \sigma \mathbb{I}_{d\times d}) > m$ |
| E | $\mathbf{X}$ is *iid* sampled from $\mathcal{U}[0,1]$ and $\mathbf{A} = \mathbf{X}\mathbf{X}^\top$ | Uniform Distribution |
| F | $\mathbf{X}$ is *iid* sampled from $\mathcal{N}(0,1)$ and $\mathbf{A} = \mathbf{X}\mathbf{X}^\top$ | Gaussian Distribution |

## 5.4 On the Invertibility of $\mathbf{W}_t^\top \mathbf{A}\mathbf{W}_t$

In the definition of the proxy matrix $\mathbf{P}_t$, there is a Moore–Penrose inverse term $(\mathbf{W}_t^\top \mathbf{A}\mathbf{W}_t)^\dagger$. In this subsection we provide a condition under which this matrix is always invertible thus the Moore–Penrose inverse $(\mathbf{W}_t^\top \mathbf{A}\mathbf{W}_t)^\dagger$ can be replaced with the matrix inverse $(\mathbf{W}_t^\top \mathbf{A}\mathbf{W}_t)^{-1}$. The reason why we care about the invertibility is that when $\mathbf{W}_t^\top \mathbf{A}\mathbf{W}_t$ is not invertible, it is rank deficient. Thus it might not be a good approximation to the high-rank covariance $\mathbf{A}$.

**Claim 5.9.** *If* $\mathrm{rank}(\mathbf{A}) \geq d - k + m$, *then, for all $t$, $\mathbf{W}_t^\top \mathbf{A}\mathbf{W}_t$ in Algorithm 2 is always invertible.*

**Remark 5.10.** *Note that the condition shown in Claim 5.9 is easy to be satisfied. Indeed, we can solve Problem (3.1) with $\mathbf{A}_\varepsilon = \mathbf{A} + \varepsilon \cdot \mathbb{I}_{d\times d}$. Thus, $\mathrm{rank}(\mathbf{A}_\varepsilon) = d \geq d - k + m$. Note that this small $\varepsilon$ perturbation on $\mathbf{A}$ does not change the optimal $\mathbf{W}$ because $\mathrm{Tr}\left(\mathbf{W}^\top \mathbf{A}_\varepsilon \mathbf{W}\right) = \mathrm{Tr}\left(\mathbf{W}^\top \mathbf{A}\mathbf{W}\right) + \varepsilon m$, which is only a constant $\varepsilon m$ added to the original objective function. Thus, the optimal $\mathbf{W}$ remains unchanged. In practice, we recommend using $\mathbf{A}_\varepsilon$ with a small $\varepsilon > 0$ to keep safe.*

# 6 Experiments

In this section, we provide experimental results to validate the effectiveness of the proposed Go and IPU on both synthetic and real-world data. In our experiments, we always use $\mathbf{A}_\varepsilon$ with $\varepsilon = 0.1$ to keep safe (Remark 5.10), except in the No. C synthetic data where we require $\mathrm{rank}(\mathbf{A}) \leq m$.

## 6.1 Synthetic Data

To show the effectiveness of the proposed method, we build a series of small-scale synthetic datasets, whose global optimum can be obtained by brute-force searching. Then we compare our methods with several state-of-the-art methods with the optimal indices and objective value in hand.

**Experiments Setup.** We compare the newly proposed Go (Algorithm 1) and IPU (Algorithm 2) with SOAP [43], SRT [47], and CSSP [27]. For the synthetic data, we fix $m = 3, k = 7$, and $d = 20$. We cannot afford large-scale setting since the brute-force searching space grows exponentially. We consider three different initialization methods: Random Subspace; Convex Relaxation proposed in [41] and used in [43]; Low Rank Approx. with $\mathrm{Go}(\mathbf{A}_m, m, k, d)$. We consider 6 different synthetic data in our experiments. The descriptions of these schemes are summarized in Table 1. For Scheme A and B, they are the synthetic data used in [43]. But we trim them to fit our setting, that is $m = 3, k = 7, d = 20$. For Scheme C, we validate the correctness that Algorithm 1 globally solves Problem (3.1). For Scheme D, we use it to see the performance comparison when the $\mathrm{rank}(\mathbf{A})$ is strictly larger than $m$. For Scheme E and F, we compare the performance when data are generated from known distribution rather than using the eigenvalues fixed covariance. For A–D, we fix the eigenvalues and generate the eigenspace randomly following [43]. Every scheme is independently run for 100 times and we report the mean and standard error. For the Random Subspace setting, every realization $\mathbf{A}$ is repeated run 20 times with different random initialization. Thus, in the random initialization setting, we run all algorithms $20 \times 100 = 2000$ times. To compute std. err. of HF, we run algorithms as we do for random initialization. The overall mean and standard error are reported.

**Performance Measures.** (1) Intersection Ratio (IR): $^{\mathrm{card}(\{\text{estimated indices}\} \cap \{\text{optimal indices}\})} / _{\#\text{ sparsity } k}$. The reason we use Intersection Ratio is that FSPCA performs feature selection and PCA simultaneously. The Intersection Ratio can measure the intersection between the indices returned by algorithm

Table 2: Synthetic Data Results. [mean (std. err.); ↑: larger is better; ↓: smaller is better]

| | | Random Subspace | | | Convex Relaxation | | | Low Rank Approx. | | |
|---|---|---|---|---|---|---|---|---|---|---|
| | | IR ↑ | RE ↓ | HF ↑ | IR ↑ | RE ↓ | HF ↑ | IR ↑ | RE ↓ | HF ↑ |
| A | SOAP | 0.73 (0.09) | 0.03 (0.02) | 0.18 (0.15) | 0.71 (0.12) | 0.08 (0.04) | 0.01 (0.01) | 0.84 (0.12) | 0.03 (0.03) | 0.22 (0.17) |
| | SRT | 0.77 (0.19) | 0.01 (0.02) | 0.70 (0.21) | 0.92 (0.12) | 0.01 (0.02) | 0.62 (0.24) | 0.88 (0.16) | 0.02 (0.04) | 0.50 (0.25) |
| | CSSP | 0.63 (0.13) | 0.88 (0.05) | 0.00 (0.00) | 0.62 (0.12) | 0.87 (0.06) | 0.00 (0.00) | 0.62 (0.12) | 0.87 (0.06) | 0.00 (0.00) |
| | Go | 0.92 (0.12) | 0.01 (0.03) | 0.74 (0.19) | 0.93 (0.12) | 0.01 (0.03) | 0.67 (0.22) | 0.93 (0.12) | 0.01 (0.03) | 0.66 (0.22) |
| | IPU | **0.97 (0.04)** | **0.00 (0.00)** | **1.00 (0.00)** | **0.99 (0.04)** | **0.00 (0.00)** | **0.97 (0.03)** | **0.98 (0.05)** | **0.00 (0.00)** | **0.91 (0.08)** |
| B | SOAP | 0.76 (0.12) | 0.03 (0.03) | 0.14 (0.12) | 0.78 (0.11) | 0.04 (0.03) | 0.09 (0.08) | 0.77 (0.12) | 0.04 (0.03) | 0.05 (0.05) |
| | SRT | 0.59 (0.08) | 0.03 (0.03) | 0.28 (0.20) | 0.79 (0.14) | 0.04 (0.04) | 0.15 (0.13) | 0.80 (0.16) | 0.04 (0.05) | 0.30 (0.21) |
| | CSSP | 0.77 (0.10) | 0.90 (0.05) | 0.00 (0.00) | 0.76 (0.12) | 0.90 (0.05) | 0.00 (0.00) | 0.76 (0.12) | 0.91 (0.05) | 0.00 (0.00) |
| | Go | **0.99 (0.02)** | **0.00 (0.00)** | **1.00 (0.00)** | **0.99 (0.02)** | **0.00 (0.00)** | **1.00 (0.00)** | **0.99 (0.01)** | **0.00 (0.00)** | **1.00 (0.00)** |
| | IPU | 0.97 (0.03) | **0.00 (0.00)** | **1.00 (0.00)** | **0.99 (0.01)** | **0.00 (0.00)** | **1.00 (0.00)** | **0.99 (0.01)** | **0.00 (0.00)** | **1.00 (0.00)** |
| C | SOAP | 0.77 (0.12) | 0.04 (0.03) | 0.11 (0.10) | 0.77 (0.12) | 0.04 (0.03) | 0.08 (0.07) | 0.76 (0.12) | 0.04 (0.03) | 0.05 (0.05) |
| | SRT | 0.59 (0.08) | 0.03 (0.04) | 0.20 (0.16) | 0.76 (0.16) | 0.05 (0.05) | 0.12 (0.11) | 0.80 (0.17) | 0.05 (0.06) | 0.26 (0.19) |
| | CSSP | 0.77 (0.11) | 0.94 (0.03) | 0.00 (0.00) | 0.76 (0.12) | 0.94 (0.03) | 0.00 (0.00) | 0.76 (0.12) | 0.94 (0.03) | 0.00 (0.00) |
| | Go | **1.00 (0.00)** | **0.00 (0.00)** | **1.00 (0.00)** | **1.00 (0.00)** | **0.00 (0.00)** | **1.00 (0.00)** | **1.00 (0.00)** | **0.00 (0.00)** | **1.00 (0.00)** |
| | IPU | **1.00 (0.00)** | **0.00 (0.00)** | **1.00 (0.00)** | **1.00 (0.00)** | **0.00 (0.00)** | **1.00 (0.00)** | **1.00 (0.00)** | **0.00 (0.00)** | **1.00 (0.00)** |
| D | SOAP | 0.79 (0.08) | 0.01 (0.01) | 0.43 (0.25) | 0.80 (0.13) | 0.02 (0.02) | 0.15 (0.13) | 0.84 (0.11) | 0.01 (0.01) | 0.22 (0.17) |
| | SRT | 0.57 (0.07) | 0.02 (0.02) | 0.14 (0.12) | 0.77 (0.15) | 0.04 (0.04) | 0.12 (0.11) | 0.83 (0.14) | 0.02 (0.03) | 0.27 (0.20) |
| | CSSP | 0.76 (0.12) | 0.80 (0.07) | 0.00 (0.00) | 0.77 (0.12) | 0.82 (0.08) | 0.00 (0.00) | 0.77 (0.12) | 0.82 (0.08) | 0.00 (0.00) |
| | Go | **0.91 (0.10)** | **0.00 (0.01)** | 0.52 (0.25) | 0.92 (0.10) | **0.00 (0.01)** | 0.59 (0.24) | **0.92 (0.09)** | **0.00 (0.01)** | 0.56 (0.25) |
| | IPU | 0.83 (0.07) | **0.00 (0.00)** | **0.97 (0.03)** | **0.93 (0.10)** | **0.00 (0.01)** | **0.65 (0.23)** | 0.92 (0.10) | **0.00 (0.01)** | **0.60 (0.24)** |
| E | SOAP | 0.43 (0.07) | 0.06 (0.03) | 0.01 (0.01) | 0.46 (0.16) | 0.12 (0.05) | 0.00 (0.00) | 0.73 (0.16) | 0.04 (0.04) | 0.12 (0.11) |
| | SRT | 0.86 (0.07) | **0.00 (0.00)** | 0.72 (0.20) | 0.88 (0.11) | 0.01 (0.01) | 0.40 (0.24) | **0.90 (0.09)** | 0.01 (0.01) | **0.52 (0.25)** |
| | CSSP | 0.43 (0.16) | 0.82 (0.06) | 0.00 (0.00) | 0.43 (0.16) | 0.83 (0.06) | 0.00 (0.00) | 0.44 (0.16) | 0.83 (0.06) | 0.00 (0.00) |
| | Go | **0.89 (0.09)** | **0.00 (0.01)** | 0.48 (0.25) | **0.90 (0.09)** | **0.00 (0.01)** | **0.46 (0.25)** | 0.88 (0.10) | **0.01(0.01)** | 0.41 (0.24) |
| | IPU | 0.83 (0.06) | **0.00 (0.00)** | **0.89 (0.10)** | 0.87 (0.10) | 0.01 (0.01) | 0.37 (0.23) | 0.88 (0.10) | **0.01(0.01)** | 0.42 (0.24) |
| F | SOAP | 0.61 (0.07) | **0.01 (0.01)** | 0.36 (0.23) | 0.79 (0.14) | **0.03 (0.03)** | 0.16 (0.13) | 0.81 (0.12) | **0.03 (0.02)** | 0.16 (0.13) |
| | SRT | 0.62 (0.08) | **0.01 (0.01)** | 0.37 (0.23) | **0.82 (0.12)** | **0.03 (0.02)** | **0.20 (0.16)** | **0.82 (0.12)** | **0.03 (0.02)** | **0.17 (0.14)** |
| | CSSP | 0.79 (0.13) | 0.52 (0.08) | 0.00 (0.00) | 0.77 (0.14) | 0.54 (0.08) | 0.00 (0.00) | 0.77 (0.14) | 0.54 (0.08) | 0.00 (0.00) |
| | Go | **0.83 (0.12)** | 0.02 (0.03) | 0.21 (0.17) | 0.81 (0.12) | **0.03 (0.03)** | 0.16 (0.13) | 0.81 (0.12) | **0.03 (0.03)** | 0.16 (0.13) |
| | IPU | 0.62 (0.07) | **0.01 (0.01)** | **0.44 (0.25)** | **0.82 (0.12)** | **0.03 (0.02)** | 0.18 (0.15) | **0.82 (0.12)** | **0.03 (0.02)** | **0.17 (0.14)** |

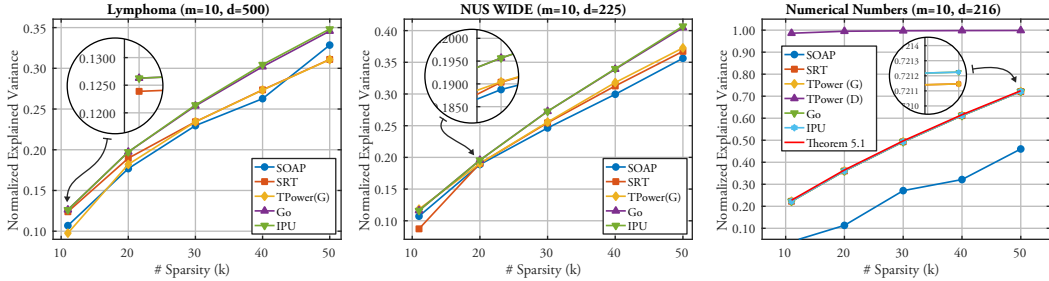

Figure 2: Real-world Data Results.

and the optimal indices. (2) Relative Error (RE): $\frac{\mathrm{Tr}\left(\mathbf{W}_*^\top \mathbf{A} \mathbf{W}_*\right) - \mathrm{Tr}\left(\mathbf{W}^\top \mathbf{A} \mathbf{W}\right)}{\mathrm{Tr}\left(\mathbf{W}_*^\top \mathbf{A} \mathbf{W}_*\right)}$. (3) Hit Frequency (HF): $\frac{1}{N}\sum_{i=1}^{N}\mathbf{1}\{\text{Relative Error} \leq 10^{-3}\}$, where $N$ is the number of repeated runs. This measure shows the frequency of the algorithm approximately reaches the global optimum.

**Results.** Experimental results are reported in Table 2, and we get the following insights: (1) From No. C, Algorithm 1 gives a globally optimal solution when the covariance $\mathbf{A}$ is low-rank. (2) Both the performance of Go and IPU outperform or match other state-of-the-art methods, especially when the numerical rank of covariance is small. (3) CSSP does not perform well in HF and RE, which is consistent with results reported in [27], since the objective of CSSP is a regression-type minimization rather than variance maximization. (4) When the Low Rank Approx. strategy (with Go) is used as initialization, all methods have match or even better explained variance than initialization with Convex Relaxation, while the computational complexity of Low Rank Approx. (with SVD) is seriously smaller than that of Convex Relaxation (with ADMM or SDP). A small but important detail: IPU is a local ascent algorithm, thus when initialized with Low Rank Approx., IPU always perform better or match than Go. Meanwhile, initialization with Random Space has better performance than both Convex Relaxation and Low Rank Approx., which is not surprising since the reported results for

Random Subspace are the maximal objective value among 20 random initialization. This strategy is widely used in practice, e.g., run $k$-means multiple times with different initialization and pick the one with the smallest loss.

## 6.2 Real-world Data

**Experiment Setup.** We consider real-world datasets, including Lymphoma (biology) [48], NUS-WIDE (web images) [10], and Numerical Numbers (handwritten numbers) [3]. We compare Go and IPU with SOAP, SRT, TPower (G) and report the results of TPower (D) as a baseline. TPower (G) selects the sparsity pattern with the leading eigenvector Greedily and TPower (D) uses the Deflation scheme, which cannot produce consistent sparsity pattern among rows. We follow [43] to use Convex Relaxation as the initialization. Following [43, 47], we use the Normalized Explained Variance as the performance measure. The Normalized Explained Variance is defined as $\text{Tr}(\widehat{\mathbf{W}}^\top \mathbf{A} \widehat{\mathbf{W}})/\text{Tr}(\mathbf{A}_m)$, where the $\widehat{\mathbf{W}}$ is the subspace estimation returned by algorithms.

**Results.** The experimental results are reported in Figure 2, from which we get the following insights: (1) For all three real-world datasets, the new algorithms, Go and IPU, consistently perform better than other state-of-the-art methods that solve FSPCA; (2) For NN dataset, the performance of all methods except SOAP and TPower (D) are tied. It is of interest to see whether the reason for this phenomenon is the dataset is too difficult or too easy. Therefore, we plot the approximation bound in Theorem 5.1, which reveals that these methods achieve almost optimal performance; (3) While TPower (D) achieves the highest NEV, it cannot be used for either feature selection or sparse subspace estimation (see Definition 1.1), due to the sparsity inconsistent issue of one-by-one eigenvectors estimation (see Figure 1). Actually, TPower (D) actually solves a less constrained problem.

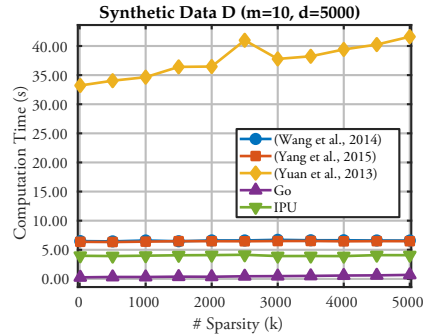

Figure 3: Computation Time.

**Computation Time.** We conducted experiments to evaluation computation time on synthetic setting D with $d = 5000, m = 10$. Please see Figure 3, which shows the new algorithms scale well for large-scale covariance. All experiments in this paper were run on MATLAB 2018a with a 2.3 GHz Quad-Core Intel Core i5 CPU and 16GB memory MBP.

**Convergence.** In Theorem 5.5, we prove the monotonic ascent property of IPU (Algorithm 2) and in Remark 5.7, we claim that existing iterative schemes are not monotonic ascent guaranteed. Here we provide numerical evidence to support this claim. We run Go, IPU, SOAP, SRT on Lymphoma dataset with $m = 10, k = 100, d = 500$. We use the same convex relaxation initialization for all methods with row truncation. We record the objective value in every iterative step for all methods. The results are plotted in Figure 4, from which we can see both SOAP and SRT are not ascent methods and both Go and IPU achieve better Explained Variance than SOAP and SRT with the same initialization. Besides, IPU takes less than 10 steps to converge, which is the case we keep seeing in all our experiments.

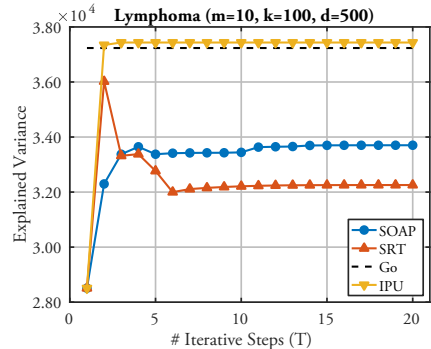

Figure 4: Convergence.

## 7 Conclusion

In this paper, we present algorithms to directly estimate the row sparsity constrained leading $m$ eigenvectors. We propose Algorithm 1 to solve FSPCA for low-rank covariance globally. For general high-rank covariance, we propose Algorithm 2 to solve FSPCA by iteratively building a carefully designed low-rank proxy covariance matrix. We prove theoretical guarantees for both algorithms on approximation and convergence. Experimental results show the promising performance of the new algorithms compared with the state-of-the-art methods.

## Broader Impact

This paper provides efficient, effective, and provable algorithms to solve the feature sparse PCA problem. The researcher who working on feature selection, dimension reduction, and graph analysis might find the techniques in this paper interesting and highly usable for real-world applications.

## Acknowledgments and Disclosure of Funding

This work was supported in part by the National Key Research and Development Program of China under Grant 2018AAA0101902, in part by the National Natural Science Foundation of China under Grant 61936014, Grant 61772427 and Grant 61751202, and in part by the Fundamental Research Funds for the Central Universities under Grant G2019KY0501.

## Footnotes

[2]If we do not insist on the eigenvalue refinement step, we can optimize the overall complexity to $O(dkmT)$ by using SVD on $\mathbf{A}\mathbf{W}(\mathbf{W}^\top \mathbf{A}\mathbf{W})^{\dagger\frac{1}{2}}$ rather than performing partial EVD on $\mathbf{A}_{\mathcal{I},\mathcal{I}}$.

[3]First, compute $\mathbf{A}\mathbf{W}$ with $O(dkm)$ since $\mathbf{W}$ is row-sparse. Then, compute $(\mathbf{W}^\top \mathbf{A}\mathbf{W})^\dagger$ with $O(km^2)$. Let the $i$th row of $\mathbf{A}\mathbf{W}$ be $[\mathbf{A}\mathbf{W}]_i$. Finally, compute the diagonal elements of $\mathbf{P}$ by $[\mathbf{A}\mathbf{W}]_i(\mathbf{W}^\top \mathbf{A}\mathbf{W})^\dagger[\mathbf{A}\mathbf{W}]_i^\top$ with $O(dm^2)$. Overall, $O(dkm)$.

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
