[Supplementary Material]

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

| | SOAP | 0.73 (0.09) | 0.03 (0.02) | 0.18 (0.15) | 0.71 (0.12) | 0.08 (0.04) | 0.01 (0.01) | 0.84 (0.12) | 0.03 (0.03) | 0.22 (0.17) |
| | SRT | 0.77 (0.19) | 0.01 (0.02) | 0.70 (0.21) | 0.92 (0.12) | 0.01 (0.02) | 0.62 (0.24) | 0.88 (0.16) | 0.02 (0.04) | 0.50 (0.25) |
| A | CSSP | 0.63 (0.13) | 0.88 (0.05) | 0.00 (0.00) | 0.62 (0.12) | 0.87 (0.06) | 0.00 (0.00) | 0.62 (0.12) | 0.87 (0.06) | 0.00 (0.00) |
| | Go | 0.92 (0.12) | 0.01 (0.03) | 0.74 (0.19) | 0.93 (0.12) | 0.01 (0.03) | 0.67 (0.22) | 0.93 (0.12) | 0.01 (0.03) | 0.66 (0.22) |
| | IPU | **0.97 (0.04)** | **0.00 (0.00)** | **1.00 (0.00)** | **0.99 (0.04)** | **0.00 (0.00)** | **0.97 (0.03)** | **0.98 (0.05)** | **0.00 (0.00)** | **0.91 (0.08)** |
| | SOAP | 0.76 (0.12) | 0.03 (0.03) | 0.14 (0.12) | 0.78 (0.11) | 0.04 (0.03) | 0.09 (0.08) | 0.77 (0.12) | 0.04 (0.03) | 0.05 (0.05) |
| | SRT | 0.59 (0.08) | 0.03 (0.03) | 0.28 (0.20) | 0.79 (0.14) | 0.04 (0.04) | 0.15 (0.13) | 0.80 (0.16) | 0.04 (0.05) | 0.30 (0.21) |
| B | CSSP | 0.77 (0.10) | 0.90 (0.05) | 0.00 (0.00) | 0.76 (0.12) | 0.90 (0.05) | 0.00 (0.00) | 0.76 (0.12) | 0.91 (0.05) | 0.00 (0.00) |
| | Go | **0.99 (0.02)** | **0.00 (0.00)** | **1.00 (0.00)** | **0.99 (0.02)** | **0.00 (0.00)** | **1.00 (0.00)** | **0.99 (0.01)** | **0.00 (0.00)** | **1.00 (0.00)** |
| | IPU | 0.97 (0.03) | **0.00 (0.00)** | **1.00 (0.00)** | **0.99 (0.01)** | **0.00 (0.00)** | **1.00 (0.00)** | **0.99 (0.01)** | **0.00 (0.00)** | **1.00 (0.00)** |
| | SOAP | 0.77 (0.12) | 0.04 (0.03) | 0.11 (0.10) | 0.77 (0.12) | 0.04 (0.03) | 0.08 (0.07) | 0.76 (0.12) | 0.04 (0.03) | 0.05 (0.05) |
| | SRT | 0.59 (0.08) | 0.03 (0.04) | 0.20 (0.16) | 0.76 (0.16) | 0.05 (0.05) | 0.12 (0.11) | 0.80 (0.17) | 0.05 (0.06) | 0.26 (0.19) |
| C | CSSP | 0.77 (0.11) | 0.94 (0.03) | 0.00 (0.00) | 0.76 (0.12) | 0.94 (0.03) | 0.00 (0.00) | 0.76 (0.12) | 0.94 (0.03) | 0.00 (0.00) |
| | Go | **1.00 (0.00)** | **0.00 (0.00)** | **1.00 (0.00)** | **1.00 (0.00)** | **0.00 (0.00)** | **1.00 (0.00)** | **1.00 (0.00)** | **0.00 (0.00)** | **1.00 (0.00)** |
| | IPU | **1.00 (0.00)** | **0.00 (0.00)** | **1.00 (0.00)** | **1.00 (0.00)** | **0.00 (0.00)** | **1.00 (0.00)** | **1.00 (0.00)** | **0.00 (0.00)** | **1.00 (0.00)** |
| | SOAP | 0.79 (0.08) | 0.01 (0.01) | 0.43 (0.25) | 0.80 (0.13) | 0.02 (0.02) | 0.15 (0.13) | 0.84 (0.11) | 0.01 (0.01) | 0.22 (0.17) |
| | SRT | 0.57 (0.07) | 0.02 (0.02) | 0.14 (0.12) | 0.77 (0.15) | 0.04 (0.04) | 0.12 (0.11) | 0.83 (0.14) | 0.02 (0.03) | 0.27 (0.20) |
| D | CSSP | 0.76 (0.12) | 0.80 (0.07) | 0.00 (0.00) | 0.77 (0.12) | 0.82 (0.08) | 0.00 (0.00) | 0.77 (0.12) | 0.82 (0.08) | 0.00 (0.00) |
| | Go | **0.91 (0.10)** | **0.00 (0.01)** | 0.52 (0.25) | 0.92 (0.10) | **0.00 (0.01)** | 0.59 (0.24) | **0.92 (0.09)** | **0.00 (0.01)** | 0.56 (0.25) |
| | IPU | 0.83 (0.07) | **0.00 (0.00)** | **0.97 (0.03)** | **0.93 (0.10)** | **0.00 (0.01)** | **0.65 (0.23)** | **0.92 (0.10)** | **0.00 (0.01)** | **0.60 (0.24)** |
| | SOAP | 0.43 (0.07) | 0.06 (0.03) | 0.01 (0.01) | 0.46 (0.16) | 0.12 (0.05) | 0.00 (0.00) | 0.73 (0.16) | 0.04 (0.04) | 0.12 (0.11) |
| | SRT | 0.86 (0.07) | **0.00 (0.00)** | 0.72 (0.20) | 0.88 (0.11) | 0.01 (0.01) | 0.40 (0.24) | **0.90 (0.09)** | 0.01 (0.01) | **0.52 (0.25)** |
| E | CSSP | 0.43 (0.16) | 0.82 (0.06) | 0.00 (0.00) | 0.43 (0.16) | 0.83 (0.06) | 0.00 (0.00) | 0.44 (0.16) | 0.83 (0.06) | 0.00 (0.00) |
| | Go | **0.89 (0.09)** | **0.00 (0.01)** | 0.48 (0.25) | **0.90 (0.09)** | **0.00 (0.01)** | **0.46 (0.25)** | 0.88 (0.10) | **0.01 (0.01)** | 0.41 (0.24) |
| | IPU | 0.83 (0.06) | **0.00 (0.00)** | **0.89 (0.10)** | 0.87 (0.10) | 0.01 (0.01) | 0.37 (0.23) | 0.88 (0.10) | **0.01 (0.01)** | 0.42 (0.24) |
| | SOAP | 0.61 (0.07) | **0.01 (0.01)** | 0.36 (0.23) | 0.79 (0.14) | **0.03 (0.03)** | 0.16 (0.13) | 0.81 (0.12) | **0.03 (0.02)** | 0.16 (0.13) |
| | SRT | 0.62 (0.08) | **0.01 (0.01)** | 0.37 (0.23) | **0.82 (0.12)** | **0.03 (0.02)** | **0.20 (0.16)** | **0.82 (0.12)** | **0.03 (0.02)** | **0.17 (0.14)** |
| F | CSSP | 0.79 (0.13) | 0.52 (0.08) | 0.00 (0.00) | 0.77 (0.14) | 0.54 (0.08) | 0.00 (0.00) | 0.77 (0.14) | 0.54 (0.08) | 0.00 (0.00) |
| | Go | **0.83 (0.12)** | 0.02 (0.03) | 0.21 (0.17) | 0.81 (0.12) | **0.03 (0.03)** | 0.16 (0.13) | 0.81 (0.12) | **0.03 (0.03)** | 0.16 (0.13) |
| | IPU | 0.62 (0.07) | **0.01 (0.01)** | **0.44 (0.25)** | **0.82 (0.12)** | **0.03 (0.02)** | 0.18 (0.15) | **0.82 (0.12)** | **0.03 (0.02)** | **0.17 (0.14)** |

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

# Appendix

In this Appendix, we provide (1) full proof of all theorems, lemmas, and corollaries; (2) discussion on the orthogonal iteration-like reformulation of Algorithm 2.

We repeat the definitions of notations for conveniences.

**Notations.**  Throughout this paper, scalars, vectors and matrices are denoted by lowercase letters, boldface lowercase letters and boldface uppercase letters, respectively; for a matrix $\mathbf{A} \in \mathbb{R}^{d \times d}$, $\mathbf{A}^\top$ denotes the transpose of $\mathbf{A}$, $\text{Tr}(\mathbf{A}) = \sum_{i=1}^{d} a_{ii}$, $\|\mathbf{A}\|_F^2 = \text{Tr}(\mathbf{A}^\top \mathbf{A})$; $\mathbf{1}_{\{\text{condition}\}}$ is the $(0, 1)$-indicator of the condition; $\mathbb{1}_n \in \mathbb{R}^n$ denotes vector with all ones; $\|\mathbf{x}\|_0$ denotes the number of non-zero elements; $\|\mathbf{W}\|_{2,0} = \sum_{i=1}^{d} \|\mathbf{w}_i\|_2^0 = \sum_{i=1}^{d} \mathbb{1}\{\|\mathbf{w}_i\| \neq 0\}$ measures the row-sparsity of $\mathbf{W}$ where $\mathbf{W} \in \mathbb{R}^{d \times m}$, $\mathbf{w}_i \in \mathbb{R}^{1 \times m}$ is the $i$th row of $\mathbf{W}$; $\mathbb{1}_{n \times n} \in \mathbb{R}^{n \times n}$ denotes the identity matrix; $\mathcal{I}(1 : k)$ is the first $k$ elements in indices $\mathcal{I}$; $\mathbf{A}^\dagger$ denotes the Moore–Penrose inverse; $\mathbf{A}_m$ is the best rank-$m$ approximation of $\mathbf{A}$ in Frobenius norm; $\text{card}(\mathcal{I})$ is the cardinality of $\mathcal{I}$; $[n] := \mathbb{Z} \cap \{i : 1 \leq i \leq n\}$. We assume that the eigenvalues $\{\lambda_i\}_{i=1}^{n}$ are arranged in descending order, i.e., $\lambda_1 \geq \lambda_2 \geq \cdots \geq \lambda_n$.

## A  Proof of Theorem 4.2

**Definition A.1** (Set of $k$th order principal submatrices). *For $m \leq k \leq d$ and matrix $\mathbf{A} \in \mathbb{R}^{d \times d}$, we define the set of $k$th order principal submatrices of $\mathbf{A}$ as*

$$\mathbb{M}_k(\mathbf{A}) = \{\mathbf{A}_{\mathcal{I},\mathcal{I}} : \mathcal{I} \subseteq [d], \mathcal{I} = \texttt{Sort}(\mathcal{I}), \text{card}(\mathcal{I}) = k \}.$$

**Observation.**  Before the formal proof, we start with an interesting observation here, which reveals the crux of our proof. When we set $k = m$ in Problem (3.1) (do not require $\text{rank}(\mathbf{A}) \leq m$), we are asking for the best $m$ features for projecting the original data into the best fit $m$ dimensional subspace. When features are independent, this setting seems reasonable. Specifically, the problem we are talking about is

$$\max_{\mathbf{W}^\top \mathbf{W} = \mathbb{1}_{m \times m}, \|\mathbf{W}\|_{2,0} \leq m} \text{Tr}\left(\mathbf{W}^\top \mathbf{A} \mathbf{W}\right).$$

Note that for each $\mathbf{W}^\top \mathbf{W} = \mathbb{1}_{m \times m}, \|\mathbf{W}\|_{2,0} \leq m$, we can rewrite it as $\mathbf{W} = \mathbf{S}\mathbf{V}$, where $\mathbf{V} \in \mathbb{R}^{m \times m}$ satisfies $\mathbf{V}^\top \mathbf{V} = \mathbb{1}_{m \times m}$ and the row selection matrix $\mathbf{S} \in \{0, 1\}^{d \times m}$ satisfies $\mathbf{S}^\top \mathbb{1}_d = \mathbb{1}_m$. It is easy to verify, for given $\mathbf{A} \in \mathbb{R}^{d \times d}$,

$$\{\mathbf{S}^\top \mathbf{A} \mathbf{S} : \mathbf{S} = \mathbb{S}_{d,m}(\texttt{Sort}(\mathcal{I})), \mathcal{I} \subseteq [d], \text{card}(\mathcal{I}) = m \} = \mathbb{M}_m(\mathbf{A}).$$

Therefore, above problem is equivalent to

$$\max_{\substack{\mathbf{V} \in \mathbb{R}^{m \times m}, \mathbf{V}^\top \mathbf{V} = \mathbb{1}_{m \times m} \\ \widetilde{\mathbf{A}} \in \mathbb{M}_m(\mathbf{A})}} \text{Tr}\left(\mathbf{V}^\top \widetilde{\mathbf{A}} \mathbf{V}\right). \tag{A.1}$$

Note that $\mathbf{V}^\top \mathbf{V} = \mathbf{V} \mathbf{V}^\top = \mathbb{1}_{m \times m}$ since $\mathbf{V}$ is square (which is not true when $k \neq m$). Combining with the fact $\text{Tr}(\mathbf{V}^\top \widetilde{\mathbf{A}} \mathbf{V}) = \text{Tr}(\widetilde{\mathbf{A}} \mathbf{V} \mathbf{V}^\top)$, Problem (A.1) can be rewritten as

$$\max_{\widetilde{\mathbf{A}} \in \mathbb{M}_m(\mathbf{A})} \text{Tr}\left(\widetilde{\mathbf{A}}\right),$$

which can be solved globally by sorting and selecting the $k$ largest diagonal elements of $\mathbf{A}$.

If we consider above argument carefully, we will realize the key point is that by setting $k = m$, we are able to write $\sum_{i=1}^{m} \lambda_i(\mathbf{S}^\top \mathbf{A} \mathbf{S})$ as $\text{Tr}(\mathbf{S}^\top \mathbf{A} \mathbf{S})$. Equivalently, $\text{rank}(\widetilde{\mathbf{A}}) = \text{rank}(\mathbf{S}^\top \mathbf{A} \mathbf{S}) \leq m$.

Note that if $\text{rank}(\mathbf{A}) \leq m$, then for all $k$ satisfies $m \leq k \leq d$, we have $\text{rank}(\widetilde{\mathbf{A}}) \leq m$ where $\widetilde{\mathbf{A}} \in \mathbb{M}_k(\mathbf{A})$. Thus, if $\text{rank}(\mathbf{A}) \leq m$, we can use the same technique (which is not using $\mathbf{V}^\top \mathbf{V} = \mathbf{V} \mathbf{V}^\top = \mathbb{1}_{m \times m}$. See detailed proof.) to solve the following problem even if $k \neq m$:

$$\max_{\substack{\mathbf{W}^\top \mathbf{W} = \mathbb{1}_{m \times m}, \|\mathbf{W}\|_{2,0} \leq k \\ \text{rank}(\mathbf{A}) \leq m}} \text{Tr}\left(\mathbf{W}^\top \mathbf{A} \mathbf{W}\right). \tag{A.2}$$

In detail, note that

$$\text{Prob. (A.2)} \Leftrightarrow \max_{\widetilde{\mathbf{A}} \in \mathbb{M}_k(\mathbf{A})} \sum_{i=1}^{k} \lambda_i(\widetilde{\mathbf{A}}) \Leftrightarrow \max_{\widetilde{\mathbf{A}} \in \mathbb{M}_k(\mathbf{A})} \text{Tr}\left(\widetilde{\mathbf{A}}\right), \tag{C}$$

which is the kernel idea of the proof of Theorem 4.2.

**Theorem 4.2.** *Suppose* $\mathbf{A} \succcurlyeq \mathbb{0}$ *and* $\text{rank}(\mathbf{A}) \leq m$. *Let* $\mathbf{W} = \text{Go}(\mathbf{A}, m, k, d)$ *with* $m \leq k \leq d$. *Then,* $\mathbf{W}$ *is a globally optimal solution of Problem* (3.1).

*Proof.* The proof is a formal argument of the equivalent chain Equation (C). Let $\mathscr{S}_{k,d}$ be the set of all $k$-from-$d$ selection matrix. Note that,

$$\max_{\mathbf{W}^\top \mathbf{W} = \mathbb{I}_{m \times m}, \|\mathbf{W}\|_{2,0} \leq k} \text{Tr}(\mathbf{W}^\top \mathbf{A} \mathbf{W})$$

$$= \max_{\mathbf{V}^\top \mathbf{V} = \mathbb{I}_{m \times m}, \mathbf{S} \in \mathscr{S}_{k,d}} \text{Tr}(\mathbf{V}^\top \mathbf{S}^\top \mathbf{A} \mathbf{S} \mathbf{V}) \qquad \text{(use } \mathbf{W} = \mathbf{S}\mathbf{V}\text{)}$$

$$= \max_{\mathbf{S} \in \mathscr{S}_{k,d}} \left( \max_{\mathbf{V}^\top \mathbf{V} = \mathbb{I}_{m \times m}} \text{Tr}(\mathbf{V}^\top \mathbf{S}^\top \mathbf{A} \mathbf{S} \mathbf{V}) \right)$$

$$= \max_{\mathbf{S} \in \mathscr{S}_{k,d}} \sum_{i=1}^{m} \lambda_i(\mathbf{S}^\top \mathbf{A} \mathbf{S}) \qquad \text{(use Ky Fan's Theorem)}$$

$$= \max_{\mathbf{S} \in \mathscr{S}_{k,d}} \sum_{i=1}^{k} \lambda_i(\mathbf{S}^\top \mathbf{A} \mathbf{S}) \qquad \text{(use } \text{rank}(\mathbf{A}) \leq m\text{)}$$

$$= \max_{\mathbf{S} \in \mathscr{S}_{k,d}} \text{Tr}(\mathbf{S}^\top \mathbf{A} \mathbf{S}) \qquad \text{(use } \mathbf{S}^\top \mathbf{A} \mathbf{S} \in \mathbb{R}^{k \times k}\text{)}$$

$$= \max_{\widetilde{\mathbf{A}} \in \mathbb{M}_k(\mathbf{A})} \text{Tr}\left(\widetilde{\mathbf{A}}\right),$$

which can be easily solved globally by first sorting the diagonal elements of $\mathbf{A}$ and selecting the $k$ largest elements then performing eigenvalue decomposition on the selected principal submatrix of $\mathbf{A}$ to obtain $\mathbf{W}$. $\qquad \square$

**Remark A.2.** *The proof of Theorem 4.2 seems clean and it seems there is no need to mention the* $k = m$ *case in the* **Observation** *paragraph. However, we insist on doing so with two reasons. One reason is that, we want to show what the Theorem 4.2 is inspired by. The other is that, this part actually gives the mildest condition under which our technique works. That is if*

$$\max_{\mathbf{S} \in \mathscr{S}_{k,d}} \sum_{i=1}^{m} \lambda_i(\mathbf{S}^\top \mathbf{A} \mathbf{S}) = \max_{\mathbf{S} \in \mathscr{S}_{k,d}} \text{Tr}(\mathbf{S}^\top \mathbf{A} \mathbf{S}), \tag{TR}$$

*holds, then Algorithm 1 gives global optimal solution. It is notable that* $\text{rank}(\mathbf{A}) \leq m$ *is a sufficient condition for TR, yet not a necessary one.*

## B    Proof of Convergence Guarantee

Before proving the ascent theorem, we first introduce some preliminary results.

**Lemma B.1** (Horn & Johnson 17, Theorem 1.3.22). *For* $\mathbf{A} \in \mathbb{R}^{n_1 \times n_2}, \mathbf{B} \in \mathbb{R}^{n_2 \times n_1}$ *with* $n_1 \leq n_2$, *we have*

$$\lambda_i(\mathbf{B}\mathbf{A}) = \begin{cases} \lambda_i(\mathbf{A}\mathbf{B}) & \textit{for} \quad 1 \leq i \leq n_1 \\ 0 & \textit{for} \quad n_1 + 1 \leq i \leq n_2. \end{cases}$$

Lemma B.1 leads to an eigenvalue estimation that will be used in our main proof.

**Corollary B.2.** *Let* $\mathbf{\Gamma} = \mathbf{X}^\top \mathbf{W}_t (\mathbf{W}_t^\top \mathbf{X}\mathbf{X}^\top \mathbf{W}_t)^\dagger \mathbf{W}_t^\top \mathbf{X}$. *For the eigenvalues of* $\mathbf{\Gamma}$, *it holds*

$$\lambda_i(\mathbf{\Gamma}) = \begin{cases} 1 & \textit{for} \quad 1 \leq i \leq r \\ 0 & \textit{for} \quad r + 1 \leq i \leq d, \end{cases}$$

*where* $r = \text{rank}(\mathbf{X}^\top \mathbf{W}_t) \leq m$.

*Proof.* Let $\mathbf{A} = (\mathbf{W}_t^\top \mathbf{X} \mathbf{X}^\top \mathbf{W}_t)^\dagger \mathbf{W}_t^\top \mathbf{X}, \mathbf{B} = \mathbf{X}^\top \mathbf{W}_t$. Thus, for each $1 \leq i \leq d$, $\lambda_i(\mathbf{\Gamma}) = \lambda_i(\mathbf{BA})$ and

$$\mathbf{AB} = (\mathbf{W}_t^\top \mathbf{X} \mathbf{X}^\top \mathbf{W}_t)^\dagger \mathbf{W}_t^\top \mathbf{X} \mathbf{X}^\top \mathbf{W}_t.$$

Using Lemma B.1, we have

$$\lambda_i(\mathbf{\Gamma}) = \begin{cases} \lambda_i(\mathbf{AB}) & \text{for} \quad 1 \leq i \leq m \\ 0 & \text{for} \quad m+1 \leq i \leq d. \end{cases}$$

Note that $\text{rank}(\mathbf{AB}) = r \leq m$ and

$$\lambda_i(\mathbf{AB}) = \begin{cases} 1 & \text{for} \quad 1 \leq i \leq r \\ 0 & \text{for} \quad r+1 \leq i \leq d. \end{cases}$$

which completes the proof. $\qquad\square$

**Lemma B.3** (Von Neumann [40]). *If matrices $\mathbf{X} \in \mathbb{R}^{n \times n}$ and $\mathbf{Y} \in \mathbb{R}^{n \times n}$ are symmetric, then,*

$$\text{Tr}(\mathbf{XY}) \leq \sum_{i=1}^n \lambda_i(\mathbf{X})\lambda_i(\mathbf{Y}).$$

*If the equality holds, $\mathbf{X}$ and $\mathbf{Y}$ are simultaneously diagonalizable.*

**Lemma B.4** (Horn & Johnson [17], Theorem 4.3.53). *If matrices $\mathbf{X} \in \mathbb{R}^{n \times n}$ and $\mathbf{Y} \in \mathbb{R}^{n \times n}$ are symmetric, then,*

$$\text{Tr}(\mathbf{XY}) \geq \sum_{i=1}^n \lambda_{n-i}(\mathbf{X})\lambda_i(\mathbf{Y}).$$

*If the equality holds, $\mathbf{X}$ and $\mathbf{Y}$ are simultaneously diagonalizable.*

Now we are ready to prove the main result which shows the objective function values generated by Algorithm 2 are monotonic ascent.

**Theorem 5.5** (Monotonic increasing). *Suppose $\mathbf{A} \succcurlyeq \mathbb{0}, m \leq k \leq d$. Let $\mathbf{W}_{t+1}$ be the variable defined in Algorithm 2. If $\mathbf{W}_t \neq \mathbf{W}_{t+1}$ up to EVD, then, $\text{Tr}(\mathbf{W}_t^\top \mathbf{A} \mathbf{W}_t) < \text{Tr}(\mathbf{W}_{t+1}^\top \mathbf{A} \mathbf{W}_{t+1})$.*

*Proof.* First, we prove $\text{Tr}(\mathbf{W}_t^\top \mathbf{A} \mathbf{W}_t) \leq \text{Tr}(\mathbf{W}_{t+1}^\top \mathbf{A} \mathbf{W}_{t+1})$. Note that

$$\begin{aligned} &\text{Tr}\left(\mathbf{W}_t^\top \mathbf{A} \mathbf{W}_t\right) \\ &\overset{\textcircled{1}}{=} \text{Tr}\left(\mathbf{W}_t^\top \mathbf{A} \mathbf{W}_t (\mathbf{W}_t^\top \mathbf{A} \mathbf{W}_t)^\dagger \mathbf{W}_t^\top \mathbf{A} \mathbf{W}_t\right) \\ &\overset{\textcircled{2}}{\leq} \text{Tr}\left(\widetilde{\mathbf{W}}_{t+1}^\top \mathbf{A} \mathbf{W}_t (\mathbf{W}_t^\top \mathbf{A} \mathbf{W}_t)^\dagger \mathbf{W}_t^\top \mathbf{A} \widetilde{\mathbf{W}}_{t+1}\right) \\ &\overset{\textcircled{3}}{=} \text{Tr}\left(\mathbf{X}^\top \mathbf{W}_t (\mathbf{W}_t^\top \mathbf{A} \mathbf{W}_t)^\dagger \mathbf{W}_t^\top \mathbf{X} \mathbf{X}^\top \widetilde{\mathbf{W}}_{t+1} \widetilde{\mathbf{W}}_{t+1}^\top \mathbf{X}\right) \end{aligned}$$

where $\textcircled{1}$ uses the fact $\mathbf{A} = \mathbf{A} \mathbf{A}^\dagger \mathbf{A}$; $\textcircled{2}$ uses $\widetilde{\mathbf{W}}_{t+1}$ maximizing Problem (3.1) for $\mathbf{P}_t$; $\textcircled{3}$ uses $\mathbf{A} \succcurlyeq \mathbb{0}$, which implies that we can always find $\mathbf{X} \in \mathbb{R}^{d \times d}$ such that $\mathbf{A} = \mathbf{X} \mathbf{X}^\top$ (e.g., with Cholesky decomposition).

Let $\mathbf{\Gamma} \in \mathbb{R}^{d \times d}, \mathbf{\Omega} \in \mathbb{R}^{d \times d}$ be

$$\begin{aligned} \mathbf{\Gamma} &= \mathbf{X}^\top \mathbf{W}_t (\mathbf{W}_t^\top \mathbf{A} \mathbf{W}_t)^\dagger \mathbf{W}_t^\top \mathbf{X} \\ \mathbf{\Omega} &= \mathbf{X}^\top \widetilde{\mathbf{W}}_{t+1} \widetilde{\mathbf{W}}_{t+1}^\top \mathbf{X}. \end{aligned}$$

Then, the RHS (right-hand side) of $\textcircled{3}$ can be rewritten as

$$\text{RHS of } \textcircled{3} = \text{Tr}(\mathbf{\Gamma}\mathbf{\Omega}) \overset{\textcircled{4}}{\leq} \sum_{i=1}^d \lambda_i(\mathbf{\Gamma})\lambda_i(\mathbf{\Omega}) \overset{\textcircled{5}}{\leq} \sum_{i=1}^m \lambda_i(\mathbf{\Omega}),$$

where $\textcircled{4}$ uses Lemma B.3; $\textcircled{5}$ uses Corollary B.2 and the fact for each $1 \leq i \leq m$, we have $\lambda_i(\mathbf{\Omega}) \geq 0$.

Note that $\text{rank}(\mathbf{\Omega}) \leq \text{rank}(\widetilde{\mathbf{W}}_{t+1}) = m$. Then we have $\sum_{i=1}^{m} \lambda_i(\mathbf{\Omega}) = \text{Tr}(\mathbf{\Omega})$. Thus the RHS of ⑤ can be rewritten as

$$\text{RHS of ⑤} = \text{Tr}(\mathbf{\Omega}) = \text{Tr}(\widetilde{\mathbf{W}}_{t+1}^{\top}\mathbf{A}\widetilde{\mathbf{W}}_{t+1}),$$

which is exactly the updated objective function value of Problem (4.1). But we can go further by notice that $\widetilde{\mathbf{W}}_{t+1} = \mathbf{S}_{t+1}\widetilde{\mathbf{V}}_{t+1}$, $\mathbf{W}_{t+1} = \mathbf{S}_{t+1}\mathbf{V}_{t+1}$, and $\mathbf{V}_{t+1}$ maximizes Problem (4.2). That gives

$$\text{Tr}(\widetilde{\mathbf{W}}_{t+1}^{\top}\mathbf{A}\widetilde{\mathbf{W}}_{t+1}) = \text{Tr}(\widetilde{\mathbf{V}}_{t+1}^{\top}\mathbf{S}_{t+1}^{\top}\mathbf{A}\mathbf{S}_{t+1}\widetilde{\mathbf{V}}_{t+1}) \leq \text{Tr}(\mathbf{W}_{t+1}^{\top}\mathbf{A}\mathbf{W}_{t+1}),$$

which proves the non-decreasing.

Then, we show that if $\mathbf{W}_t \neq \mathbf{W}_{t+1}$ up to EVD, the equality cannot hold, which indicates strictly increasing. The proof is by contradiction.

Suppose there exists $\mathbf{W}_t \neq \mathbf{W}_{t+1}$ up to EVD such that $\text{Tr}(\mathbf{W}_t^{\top}\mathbf{A}\mathbf{W}_t) = \text{Tr}(\mathbf{W}_{t+1}^{\top}\mathbf{A}\mathbf{W}_{t+1})$ and we write $\mathbf{W}_t = \mathbf{S}_t\mathbf{V}_t$, $\mathbf{W}_{t+1} = \mathbf{S}_{t+1}\mathbf{V}_{t+1}$. Then, it should have $\mathbf{S}_t \neq \mathbf{S}_{t+1}$. That is because $\mathbf{V}_t, \mathbf{V}_{t+1}$ are the top eigenvectors of $\mathbf{S}_t^{\top}\mathbf{A}\mathbf{S}_t, \mathbf{S}_{t+1}^{\top}\mathbf{A}\mathbf{S}_{t+1}$, respectively. And if $\mathbf{S}_t = \mathbf{S}_{t+1}$, it must hold $\mathbf{W}_t = \mathbf{W}_{t+1}$ up to EVD, contradiction.

In the sequel, we assume $\mathbf{S}_t \neq \mathbf{S}_{t+1}$. Note that ④ is equality now. Using the condition of equality of Lemma B.3, we have $\mathbf{X}^{\top}\mathbf{W}_t(\mathbf{W}_t^{\top}\mathbf{A}\mathbf{W}_t)^{\dagger}\mathbf{W}_t^{\top}\mathbf{X}$ and $\mathbf{X}^{\top}\mathbf{W}_{t+1}\mathbf{W}_{t+1}^{\top}\mathbf{X}$ are simultaneously diagonalizable. From now on, without loss of generality, we assume $\mathbf{A}$ is full rank. Then, we have $\mathbf{X}^{\top}\mathbf{W}_t(\mathbf{W}_t^{\top}\mathbf{A}\mathbf{W}_t)^{\dagger}\mathbf{W}_t^{\top}\mathbf{X} = \mathbf{X}^{\top}\mathbf{W}_t\mathbf{\Sigma}_t^{-1}\mathbf{W}_t^{\top}\mathbf{X}$ where $\mathbf{\Sigma}_t = \mathbf{W}_t^{\top}\mathbf{A}\mathbf{W}_t$ is diagonal. Let $\mathbf{Q}_t = \mathbf{X}^{\top}\mathbf{W}_t\mathbf{\Sigma}_t^{-1/2}$. It is easy to check $\mathbf{X}^{\top}\mathbf{W}_t(\mathbf{W}_t^{\top}\mathbf{A}\mathbf{W}_t)^{\dagger}\mathbf{W}_t^{\top}\mathbf{X} = \mathbf{Q}_t\mathbf{Q}_t^{\top}$ and $\mathbf{Q}_t^{\top}\mathbf{Q}_t = \mathbf{\Sigma}_t^{-1/2}\mathbf{W}_t^{\top}\mathbf{A}\mathbf{W}_t\mathbf{\Sigma}_t^{-1/2} = \mathbf{\Sigma}_t^{-1/2}\mathbf{\Sigma}_t\mathbf{\Sigma}_t^{-1/2} = \mathbb{I}_{m \times m}$. Using the simultaneously diagonalizable property and Lemma B.1, we have

$$\mathbf{X}^{\top}\mathbf{W}_{t+1}\mathbf{W}_{t+1}^{\top}\mathbf{X} = \mathbf{Q}_t\mathbf{\Sigma}_{t+1}\mathbf{Q}_t^{\top} = \mathbf{X}^{\top}\mathbf{W}_t\mathbf{\Sigma}_t^{-1/2}\mathbf{\Sigma}_{t+1}\mathbf{\Sigma}_t^{-1/2}\mathbf{W}_t^{\top}\mathbf{X}.$$

which gives

$$\begin{aligned}
&\mathbf{W}_t^{\top}\mathbf{A}\mathbf{W}_{t+1}\mathbf{W}_{t+1}^{\top}\mathbf{A}\mathbf{W}_t \\
=&\mathbf{W}_t^{\top}\mathbf{X}\mathbf{X}^{\top}\mathbf{W}_{t+1}\mathbf{W}_{t+1}^{\top}\mathbf{X}\mathbf{X}^{\top}\mathbf{W}_t \\
=&\mathbf{W}_t^{\top}\mathbf{X}\mathbf{X}^{\top}\mathbf{W}_t\mathbf{\Sigma}_t^{-1/2}\mathbf{\Sigma}_{t+1}\mathbf{\Sigma}_t^{-1/2}\mathbf{W}_t^{\top}\mathbf{X}\mathbf{X}^{\top}\mathbf{W}_t \\
=&\mathbf{\Sigma}_t\mathbf{\Sigma}_{t+1}.
\end{aligned}$$

Then, we consider $\text{Tr}(\mathbf{W}_t^{\top}\mathbf{P}_{t+1}\mathbf{W}_t) = \text{Tr}(\mathbf{W}_t^{\top}\mathbf{A}\mathbf{W}_{t+1}\mathbf{\Sigma}_{t+1}^{-1}\mathbf{W}_{t+1}^{\top}\mathbf{A}\mathbf{W}_t)$. Using the decomposition in ③ and Lemma B.3, we have

$$\text{Tr}(\mathbf{W}_t^{\top}\mathbf{P}_{t+1}\mathbf{W}_t) \leq \text{Tr}(\mathbf{W}_t^{\top}\mathbf{A}\mathbf{W}_t) = \text{Tr}(\mathbf{\Sigma}_t).$$

Let $\mathbf{\Pi} \in \mathbb{R}^{m \times m}, \mathbf{\Psi} \in \mathbb{R}^{m \times m}$ be

$$\begin{aligned}
\mathbf{\Pi} =&\mathbf{\Sigma}_{t+1}^{-1} \\
\mathbf{\Psi} =&\mathbf{W}_{t+1}^{\top}\mathbf{A}\mathbf{W}_t\mathbf{W}_t^{\top}\mathbf{A}\mathbf{W}_{t+1}.
\end{aligned}$$

Then, we have

$$\text{Tr}(\mathbf{W}_t^{\top}\mathbf{P}_{t+1}\mathbf{W}_t) = \text{Tr}(\mathbf{\Pi}\mathbf{\Psi}) \overset{⑥}{\geq} \sum_{i=1}^{m} \lambda_{m-i}(\mathbf{\Pi})\lambda_i(\mathbf{\Psi}) \overset{⑦}{=} \sum_{i=1}^{m} \frac{\lambda_i(\mathbf{\Sigma}_t\mathbf{\Sigma}_{t+1})}{\lambda_i(\mathbf{\Sigma}_{t+1})} = \text{Tr}(\mathbf{\Sigma}_t),$$

where ⑥ is by Lemma B.4 and ⑦ is using Lemma B.1, $\lambda_{m-i}(\mathbf{\Pi}) = \lambda_i(\mathbf{\Sigma}_{t+1})^{-1}$, and $\mathbf{W}_t^{\top}\mathbf{A}\mathbf{W}_{t+1}\mathbf{W}_{t+1}^{\top}\mathbf{A}\mathbf{W}_t = \mathbf{\Sigma}_t\mathbf{\Sigma}_{t+1}$. Combining with $\text{Tr}(\mathbf{W}_t^{\top}\mathbf{P}_{t+1}\mathbf{W}_t) \leq \text{Tr}(\mathbf{\Sigma}_t)$, the equality in ⑥ holds. Using the condition of equality of Lemma B.4, $\mathbf{\Pi}$ and $\mathbf{\Psi}$ are simultaneously diagonalizable, which indicates $\mathbf{\Psi} = \mathbf{W}_{t+1}^{\top}\mathbf{A}\mathbf{W}_t\mathbf{W}_t^{\top}\mathbf{A}\mathbf{W}_{t+1} = \mathbf{\Sigma}_t\mathbf{\Sigma}_{t+1}$. Thus $\mathbf{W}_{t+1}^{\top}\mathbf{A}\mathbf{W}_t = \mathbf{\Sigma}_t^{1/2}\mathbf{\Sigma}_{t+1}^{1/2}$ is diagonal.

Let $\mathbf{T}_t = \mathbf{X}^{\top}\mathbf{W}_t\mathbf{\Sigma}_t^{-1/2}, \mathbf{T}_{t+1} = \mathbf{X}^{\top}\mathbf{W}_{t+1}\mathbf{\Sigma}_{t+1}^{-1/2}$. It is easy to check

$$\begin{aligned}
\mathbf{T}_t^{\top}\mathbf{T}_t =&\mathbf{\Sigma}_t^{-1/2}\mathbf{W}_t^{\top}\mathbf{A}\mathbf{W}_t\mathbf{\Sigma}_t^{-1/2} = \mathbb{I}_{m \times m} \\
\mathbf{T}_{t+1}^{\top}\mathbf{T}_{t+1} =&\mathbf{\Sigma}_{t+1}^{-1/2}\mathbf{W}_{t+1}^{\top}\mathbf{A}\mathbf{W}_{t+1}\mathbf{\Sigma}_{t+1}^{-1/2} = \mathbb{I}_{m \times m} \\
\mathbf{T}_{t+1}^{\top}\mathbf{T}_t =&\mathbf{\Sigma}_{t+1}^{-1/2}\mathbf{W}_{t+1}^{\top}\mathbf{A}\mathbf{W}_t\mathbf{\Sigma}_t^{-1/2} = \mathbb{I}_{m \times m}.
\end{aligned}$$

Therefore, $\mathbf{T}_t = \mathbf{T}_{t+1}$, which indicates $\mathbf{AW}_t \boldsymbol{\Sigma}_t^{-1/2} = \mathbf{AW}_{t+1} \boldsymbol{\Sigma}_{t+1}^{-1/2}$. As $\mathbf{A}$ is full rank matrix, we conclude that $\mathbf{W}_t \boldsymbol{\Sigma}_t^{-1/2} = \mathbf{W}_{t+1} \boldsymbol{\Sigma}_{t+1}^{-1/2}$. Note that right multiplication does not change the sparsity pattern of $\mathbf{W}_t$ and $\mathbf{W}_{t+1}$. Thus, $\mathbf{S}_t = \mathbf{S}_{t+1}$, contradiction.

The proof completes. $\qquad\square$

**Theorem 5.8** (Convergence). *Suppose $\mathbf{A} \succcurlyeq \mathbb{0}, m \leq k \leq d$, and $\lambda_m - \lambda_{m+1} > 0$ on the selected principal submatrix of fixed point. Let $\{\mathbf{W}_t\}_{t=1}^\infty$ be any sequence generated by Algorithm 2. Then, the sequence $\{\mathbf{W}_t\}_{t=1}^\infty$ converges to a fixed point, say $\widetilde{\mathbf{W}}$, of Algorithm 2 in the sense of subspace, and $\|\sin\Theta\left(\operatorname{span}(\mathbf{W}_{t+1}), \operatorname{span}(\mathbf{W}_t)\right)\|_2 \to 0, \operatorname{Tr}(\mathbf{W}_t^\top \mathbf{AW}_t) \to \operatorname{Tr}(\widetilde{\mathbf{W}}^\top \mathbf{A}\widetilde{\mathbf{W}})$.*

*Proof.* Note that $\operatorname{Tr}(\mathbf{W}_t^\top \mathbf{AW}_t)$ is upper bounded by $\operatorname{Tr}(\mathbf{A})$. By Theorem 5.5, the sequence of objective function value $\{\operatorname{Tr}(\mathbf{W}_t^\top \mathbf{AW}_t)\}_{t=1}^\infty$ is strictly increasing and thus convergent. If the objective function value converges, then $\operatorname{Tr}(\mathbf{W}_t^\top \mathbf{AW}_t) = \operatorname{Tr}(\mathbf{W}_{t+1}^\top \mathbf{AW}_{t+1})$. Using the contrapositive of Theorem 5.5, it holds $\mathbf{W}_t = \mathbf{W}_{t+1}$ up to EVD, that is to say, $\mathbf{S}_t = \mathbf{S}_{t+1}$. As the eigenspace is well-defined by the eigengap assumption, it holds $\|\sin\Theta\left(\operatorname{span}(\mathbf{W}_{t+1}), \operatorname{span}(\mathbf{W}_t)\right)\|_2 = 0$. $\qquad\square$

## C  Proof of Claim 4.5

**Claim 4.5.** *For each $t \geq 1$, $\mathbf{W}_t^\top \mathbf{W}_t = \mathbb{I}_{m \times m}$, it holds $\operatorname{rank}(\mathbf{P}_t) \leq m$, and $\mathbf{P}_t \succcurlyeq \mathbb{0}$.*

*Proof.* The first part is from $\operatorname{rank}(\mathbf{P}_t) \leq \operatorname{rank}(\mathbf{W}_t) = m$. Let $\boldsymbol{\Phi} = \mathbf{AW}_t (\mathbf{W}_t^\top \mathbf{AW}_t)^\dagger \mathbf{W}_t^\top \mathbf{X}$. Using the facts $\mathbf{A} \succcurlyeq \mathbb{0}$ (which implies $\mathbf{A} = \mathbf{XX}^\top$ with Cholesky) and $\mathbf{B}^\dagger = \mathbf{B}^\dagger \mathbf{BB}^\dagger$, the second part is from

$$\mathbf{P}_t = \mathbf{AW}_t (\mathbf{W}_t^\top \mathbf{AW}_t)^\dagger \mathbf{W}_t^\top \mathbf{XX}^\top \mathbf{W}_t (\mathbf{W}_t^\top \mathbf{AW}_t)^\dagger \mathbf{W}_t^\top \mathbf{A} = \boldsymbol{\Phi}\boldsymbol{\Phi}^\top \succcurlyeq 0,$$

which completes the proof. $\qquad\square$

## D  Proof of Claim 5.9

First of all, we need a result to bound the eigenvalues of principal submatrix.

**Lemma D.1** (Horn & Johnson 17, Theorem 4.3.28). *Let $\mathbf{A} \in \mathbb{R}^{d \times d}$ be symmetric matrix that can be partitioned as*

$$\mathbf{A} = \begin{bmatrix} \mathbf{B} & \mathbf{C} \\ \mathbf{C}^\top & \mathbf{D} \end{bmatrix},$$

*where $\mathbf{B} \in \mathbb{R}^{k \times k}, \mathbf{C} \in \mathbb{R}^{k \times (d-k)}, \mathbf{D} \in \mathbb{R}^{(d-k) \times (d-k)}$. Let the eigenvalues of $\mathbf{A}$ and $\mathbf{B}$ be sorted in descending order. Then, for each $1 \leq i \leq k$, we have $\lambda_i(\mathbf{A}) \geq \lambda_i(\mathbf{B}) \geq \lambda_{d-k+i}(\mathbf{A})$.*

**Claim 5.9.** *If $\operatorname{rank}(\mathbf{A}) \geq d - k + m$, then, for all $t$, $\mathbf{W}_t^\top \mathbf{AW}_t$ in Algorithm 2 is always invertible.*

*Proof.* Note that $\mathbf{W}_t = \mathbf{S}_t \mathbf{V}_t$, where $\mathbf{S}_t = \mathbb{S}_{d,k}(\mathcal{I})$. Let $\widetilde{\mathbf{S}}_t = \mathbb{S}_{d,k}([d] \setminus \mathcal{I})$. It is easy to check that there exists a permutation matrix $\mathbf{U} = [\mathbf{S}_t \ \widetilde{\mathbf{S}}_t]$ such that

$$\mathbf{U}^\top \mathbf{AU} = \begin{bmatrix} \mathbf{S}_t^\top \mathbf{AS}_t & \mathbf{S}^\top \mathbf{A}\widetilde{\mathbf{S}}_t \\ \widetilde{\mathbf{S}}_t^\top \mathbf{AS}_t & \widetilde{\mathbf{S}}_t^\top \mathbf{A}\widetilde{\mathbf{S}}_t \end{bmatrix}.$$

Note that for each $1 \leq i \leq d$, we have $\lambda_i(\mathbf{U}^\top \mathbf{AU}) = \lambda_i(\mathbf{A})$ since $\mathbf{U}$ is permutation. Using Lemma D.1, we have for each $1 \leq i \leq m$,

$$\lambda_i(\mathbf{W}_t^\top \mathbf{AW}_t) = \lambda_i(\mathbf{S}_t^\top \mathbf{AS}_t) \geq \lambda_{d-k+i}(\mathbf{U}^\top \mathbf{AU}) = \lambda_{d-k+i}(\mathbf{A}).$$

Using $\operatorname{rank}(\mathbf{A}) \geq d - k + m$, the proof completes. $\qquad\square$

# E  Proof of Approximation Guarantee

**Theorem 5.1.** *Suppose* $\mathbf{A} \succcurlyeq \mathbb{0}$ *with condition number* $\kappa$, $m \leq k \leq d$. *Let* $\mathbf{W}_m = Go(\mathbf{A}_m, m, k, d)$, *and* $\mathbf{W}_*$ *be globally optimal for Problem 3.1. Then, we have* $(1 - \varepsilon) \leq \frac{\mathrm{Tr}(\mathbf{W}_m^\top \mathbf{A} \mathbf{W}_m)}{\mathrm{Tr}(\mathbf{W}_*^\top \mathbf{A} \mathbf{W}_*)} \leq 1$ *with*

$$\varepsilon \leq \min \left\{ \frac{dG_1}{k}, \frac{dG_2}{m}, 1 - \kappa^{-1}, 1 - \frac{k}{d} \right\}.$$

*Proof.* We first show $\varepsilon \leq \min\{\frac{dG_1}{k}, \frac{dG_2}{m}\}$. Let $\mathbf{A}_m^c = \mathbf{A} - \mathbf{A}_m$. Note that

$$
\begin{aligned}
& \mathrm{Tr}(\mathbf{W}_*^\top \mathbf{A} \mathbf{W}_*) \\
={} & \max_{\mathbf{W}^\top \mathbf{W} = \mathbb{I}_{m \times m}, \|\mathbf{W}\|_{2,0} \leq k} \mathrm{Tr}(\mathbf{W}^\top \mathbf{A} \mathbf{W}) \\
={} & \max_{\mathbf{W}^\top \mathbf{W} = \mathbb{I}_{m \times m}, \|\mathbf{W}\|_{2,0} \leq k} \mathrm{Tr}(\mathbf{W}^\top \mathbf{A}_m \mathbf{W}) + \mathrm{Tr}(\mathbf{W}^\top \mathbf{A}_m^c \mathbf{W}) \\
\leq{} & \max_{\mathbf{W}^\top \mathbf{W} = \mathbb{I}_{m \times m}, \|\mathbf{W}\|_{2,0} \leq k} \mathrm{Tr}(\mathbf{W}^\top \mathbf{A}_m \mathbf{W}) + \max_{\mathbf{W}^\top \mathbf{W} = \mathbb{I}_{m \times m}, \|\mathbf{W}\|_{2,0} \leq k} \mathrm{Tr}(\mathbf{W}^\top \mathbf{A}_m^c \mathbf{W}) \\
\leq{} & \mathrm{Tr}(\mathbf{W}_m^\top \mathbf{A}_m \mathbf{W}_m) + \max_{\mathbf{W}^\top \mathbf{W} = \mathbb{I}_{m \times m}} \mathrm{Tr}(\mathbf{W}^\top \mathbf{A}_m^c \mathbf{W}) \\
\leq{} & \mathrm{Tr}(\mathbf{W}_m^\top \mathbf{A}_m \mathbf{W}_m) + \sum_{i=m+1}^{2m} \lambda_i(\mathbf{A}),
\end{aligned}
$$

which indicates

$$\frac{\mathrm{Tr}(\mathbf{W}_m^\top \mathbf{A}_m \mathbf{W}_m)}{\mathrm{Tr}(\mathbf{W}_*^\top \mathbf{A} \mathbf{W}_*)} \geq 1 - \frac{\sum_{i=m+1}^{2m} \lambda_i(\mathbf{A})}{\mathrm{Tr}(\mathbf{W}_*^\top \mathbf{A} \mathbf{W}_*)}.$$

Note that,

$$\mathrm{Tr}(\mathbf{W}_*^\top \mathbf{A} \mathbf{W}_*) \geq \max_{\mathbf{W}^\top \mathbf{W} = \mathbb{I}_{m \times m}, \|\mathbf{W}\|_{2,0} \leq m} \mathrm{Tr}(\mathbf{W}^\top \mathbf{A} \mathbf{W}) \geq \frac{m}{d} \mathrm{Tr}(\mathbf{A}) = \frac{m}{d} \sum_{i=1}^{d} \lambda_i(\mathbf{A}),$$

where the first inequality uses $m \leq k$ and the second inequality uses Theorem 4.2. Besides,

$$\mathrm{Tr}(\mathbf{W}_*^\top \mathbf{A} \mathbf{W}_*) \geq \max_{\mathbf{W}^\top \mathbf{W} = \mathbb{I}_{m \times m}, \|\mathbf{W}\|_{2,0} \leq k} \mathrm{Tr}(\mathbf{W}^\top \mathbf{A}_m \mathbf{W}) \geq \frac{k}{d} \sum_{i=1}^{d} \lambda_i(\mathbf{A}_m) = \frac{k}{d} \sum_{i=1}^{m} \lambda_i(\mathbf{A}),$$

where the first inequality uses $\mathbf{A} \succcurlyeq \mathbb{0}$ and the second inequality uses Theorem 4.2.

Let $r = \min\{\mathrm{rank}(\mathbf{A}), 2m\}$, and

$$G_1 = \frac{\sum_{i=m+1}^{r} \lambda_i(\mathbf{A})}{\sum_{i=1}^{m} \lambda_i(\mathbf{A})}, \qquad G_2 = \frac{\sum_{i=m+1}^{r} \lambda_i(\mathbf{A})}{\sum_{i=1}^{d} \lambda_i(\mathbf{A})}.$$

Thus, we have

$$1 \geq \frac{\mathrm{Tr}(\mathbf{W}_m^\top \mathbf{A}_m \mathbf{W}_m)}{\mathrm{Tr}(\mathbf{W}_*^\top \mathbf{A} \mathbf{W}_*)} \geq 1 - \min \left\{ \frac{dG_1}{k}, \frac{dG_1}{m} \right\}.$$

Because $\mathbf{A} \succcurlyeq \mathbb{0}$, we have

$$\mathrm{Tr}(\mathbf{W}_m^\top \mathbf{A} \mathbf{W}_m) = \mathrm{Tr}(\mathbf{W}_m^\top \mathbf{A}_m \mathbf{W}_m) + \mathrm{Tr}(\mathbf{W}_m^\top \mathbf{A}_m^c \mathbf{W}_m) \geq \mathrm{Tr}(\mathbf{W}_m^\top \mathbf{A}_m \mathbf{W}_m),$$

which shows the first claim.

To show $\varepsilon \leq 1 - \kappa^{-1}$, we lower bound the objective value of $\mathbf{W}_m$ by using the Poincaré separation theorem in Lemma D.1 and the sparsity encoding $\mathbf{W} = \mathbf{SV}$,

$$\mathrm{Tr}(\mathbf{W}_m^\top \mathbf{A} \mathbf{W}_m) \geq \min_{\mathbf{S} \in \mathscr{S}_{k,d}} \max_{\mathbf{V}^\top \mathbf{V} = \mathbb{I}_{m \times m}} \mathrm{Tr}(\mathbf{V}^\top \mathbf{S}^\top \mathbf{A} \mathbf{S} \mathbf{V}) \geq \sum_{i=d-m+1}^{d} \lambda_i \geq m \cdot \lambda_d,$$

where $\mathscr{S}_{k,d}$ is the set of all $k$-from-$d$ selection matrices used in the proof of Theorem 4.2. The upper bound of the optimal objective value is by Ky Fan's Theorem:

$$\mathrm{Tr}(\mathbf{W}_*^\top \mathbf{A} \mathbf{W}_*) \leq \max_{\mathbf{W}^\top \mathbf{W} = \mathbb{I}_{m \times m}} \mathrm{Tr}(\mathbf{W}^\top \mathbf{A} \mathbf{W}) = \sum_{i=1}^{m} \lambda_i \leq m \cdot \lambda_1 = m \cdot \kappa \lambda_d.$$

Meanwhile, $\varepsilon \leq 1 - kd^{-1}$ holds by using

$$\mathrm{Tr}(\mathbf{W}_m^\top \mathbf{A} \mathbf{W}_m) \geq \mathrm{Tr}(\mathbf{W}_m^\top \mathbf{A}_m \mathbf{W}_m) \geq \frac{k}{d} \mathrm{Tr}(\mathbf{A}_m) \geq \frac{k}{d} \sum_{i=1}^{m} \lambda_i,$$

and

$$\mathrm{Tr}(\mathbf{W}_*^\top \mathbf{A} \mathbf{W}_*) \leq \sum_{i=1}^{m} \lambda_i,$$

which completes the proof. $\qquad\square$

# F Proof of Exponential Distribution Corollary 5.3

**Corollary 5.3** (Exponential distribution). *Suppose $\mathbf{A} \succcurlyeq \mathbb{0}, m \leq k \leq d$, and $\lambda_i(\mathbf{A}) = c'e^{-ci}$ with $c' > 0, c > 0$ for each $i = 1, \ldots, 2m$. Let $\mathbf{W}_m = Go(\mathbf{A}_m, m, k, d)$, and $\mathbf{W}_*$ be an optimal solution of Problem 3.1. If $m \geq \Omega\left(\frac{1}{c}\log\left(\frac{d}{k\varepsilon}\right)\right)$, then we have $(1 - \varepsilon) \leq \frac{\mathrm{Tr}(\mathbf{W}_m^\top \mathbf{A} \mathbf{W}_m)}{\mathrm{Tr}(\mathbf{W}_*^\top \mathbf{A} \mathbf{W}_*)} \leq 1$.*

*Proof.* We calculate the $G_1$ and then use the Theorem 5.1. For $G_1$, we have

$$G_1 = \frac{\sum_{i=m+1}^{r} \lambda_i(\mathbf{A})}{\sum_{i=1}^{m} \lambda_i(\mathbf{A})} = \frac{\sum_{i=m+1}^{2m} c'e^{-ci}}{\sum_{i=1}^{m} c'e^{-ci}} = \frac{\frac{e^{-c(m+1)}(1-e^{-cm})}{1-e^{-c}}}{\frac{e^{-c}(1-e^{-cm})}{1-e^{-c}}} = e^{-cm}.$$

Plug the $G_1$ into Theorem 5.1 and we have if

$$m \geq \Omega\left(\frac{1}{c}\log\left(\frac{d}{k\varepsilon}\right)\right),$$

then the approximation ratio $\varepsilon$ holds, which completes the proof. $\qquad\square$

# G Proof of Zipf-like Corollary 5.4

To prove the corollary, we need the following auxiliary lemmas.

**Lemma G.1.** *For $a, b \in \mathbb{N}, a \leq b, t \geq 1$, it holds*

$$\frac{(b+1)^{1-t} - a^{1-t}}{1-t} \leq \sum_{i=a}^{b} \frac{1}{i^t} \leq \frac{b^{1-t} - (a-1)^{1-t}}{1-t}.$$

*Proof.* Using approximation by definite integrals, we have

$$\sum_{i=a}^{b} \frac{1}{i^t} \leq \int_{a-1}^{b} \frac{1}{i^t}\mathrm{d}i = \left.\frac{i^{1-t}}{1-t}\right|_{a-1}^{b} = \frac{b^{1-t} - (a-1)^{1-t}}{1-t},$$

$$\sum_{i=a}^{b} \frac{1}{i^t} \geq \int_{a}^{b+1} \frac{1}{i^t}\mathrm{d}i = \left.\frac{i^{1-t}}{1-t}\right|_{a}^{b+1} = \frac{(b+1)^{1-t} - a^{1-t}}{1-t}$$

which completes the proof. $\qquad\square$

**Lemma G.2.** *For $t \geq 1$, it holds*

$$\frac{\sum_{i=m+1}^{2m} \frac{1}{i^t}}{\sum_{i=1}^{m} \frac{1}{i^t}} \leq \frac{1}{2^t}.$$

*Proof.* Note that,

$$\sum_{i=1}^{m} \frac{1}{i^t} - 2^t \sum_{i=m+1}^{2m} \frac{1}{i^t} = \sum_{i=1}^{m} \frac{1}{i^t} - \sum_{i=m+1}^{2m} \frac{1}{(i/2)^t} = \sum_{i=1}^{m} \left( \frac{1}{i^t} - \frac{1}{\left(\frac{i+m}{2}\right)^t} \right) \geq 0,$$

which completes the proof. $\square$

**Lemma G.3** (Bound $G_1$).

$$G_1 = \frac{\sum_{i=m+1}^{r} \lambda_i(\mathbf{A})}{\sum_{i=1}^{m} \lambda_i(\mathbf{A})} = \frac{\sum_{i=m+1}^{2m} \frac{1}{i^t}}{\sum_{i=1}^{m} \frac{1}{i^t}} \leq \min \left\{ \frac{1}{m^{t-1}}, \frac{1}{2^t} \right\}.$$

*Proof.* Using Lemma G.2, we have

$$\frac{\sum_{i=m+1}^{2m} \frac{1}{i^t}}{\sum_{i=1}^{m} \frac{1}{i^t}} \leq \frac{1}{2^t}.$$

Leveraging the Lemma G.1, we have

$$\sum_{i=m+1}^{2m} \frac{1}{i^t} \leq \frac{m^{1-t} - (2m)^{1-t}}{t-1}, \quad \sum_{i=1}^{m} \frac{1}{i^t} \geq \frac{1 - (m+1)^{1-t}}{t-1},$$

which gives

$$\frac{\sum_{i=m+1}^{2m} \frac{1}{i^t}}{\sum_{i=1}^{m} \frac{1}{i^t}} \leq \frac{m^{1-t} - (2m)^{1-t}}{1 - (m+1)^{1-t}} = \frac{\frac{1}{m^{t-1}} \left(1 - \frac{1}{2^{t-1}}\right)}{1 - \frac{1}{(m+1)^{t-1}}} = \frac{1 - \frac{1}{2^{t-1}}}{m^{t-1} \left(1 - \frac{1}{(m+1)^{t-1}}\right)}.$$

Note that $m \geq 1$, which implies

$$\frac{\sum_{i=m+1}^{2m} \frac{1}{i^t}}{\sum_{i=1}^{m} \frac{1}{i^t}} \leq \frac{1 - \frac{1}{2^{t-1}}}{m^{t-1} \left(1 - \frac{1}{2^{t-1}}\right)} \leq \frac{1}{m^{t-1}}.$$

The proof completes. $\square$

**Corollary 5.4** (Zipf's distribution). *Suppose $\mathbf{A} \succcurlyeq 0, m \leq k \leq d$, and $\lambda_i(\mathbf{A}) = ci^{-t}$ with $t > 1, c > 0$ for each $i = 1, \ldots, 2m$. Let $\mathbf{W}_m = Go(\mathbf{A}_m, m, k, d)$, and $\mathbf{W}_*$ be an optimal solution of Problem 3.1. If $m \geq \Omega \left( \left(\frac{d}{k\varepsilon}\right)^{\frac{1}{t-1}} \right)$, then we have $(1 - \varepsilon) \leq \frac{\mathrm{Tr}(\mathbf{W}_m^\top \mathbf{A} \mathbf{W}_m)}{\mathrm{Tr}(\mathbf{W}_*^\top \mathbf{A} \mathbf{W}_*)} \leq 1$.*

*Proof.* Using Lemma G.3, we have

$$G_1 = \frac{\sum_{i=m+1}^{r} \lambda_i(\mathbf{A})}{\sum_{i=1}^{m} \lambda_i(\mathbf{A})} \leq \frac{\sum_{i=m+1}^{2m} \frac{1}{i^t}}{\sum_{i=1}^{m} \frac{1}{i^t}} \leq \min \left\{ \frac{1}{m^{t-1}}, \frac{1}{2^t} \right\}.$$

Plug the $G_1$ into Theorem 5.1 and we have if

$$m \geq \Omega \left( \left(\frac{d}{k\varepsilon}\right)^{\frac{1}{t-1}} \right),$$

then the approximation ratio $\varepsilon$ holds, which completes the proof. $\square$

# H   Approximation Ratio on Real-world Data

In this section, we compute the $\varepsilon$ in the approximation ratio of Theorem 5.1 on the real-world data used in Section 6. The following figure show the approximation bound in Theorem 5.1 is not vacuous and provides useful certification on the quality of the solution.

Figure 5: Approximation Ratio in Theorem 5.1 on Real-world Data.

# I    Orthogonal Iteration-like Reformulation of Algorithm 2

We provide an orthogonal iteration-like reformulation of Algorithm 2.

Let $\mathbf{Q} = \mathbf{A}^{\frac{1}{2}}\mathbf{W}(\mathbf{W}^\top\mathbf{A}\mathbf{W})^{\frac{1}{2}\dagger}$. It is easy to see $\mathbf{Q}$ is the orthonormalization of $\mathbf{A}^{\frac{1}{2}}\mathbf{W}$ and can be computed efficiently with, e.g., the Gram-Schmidt process. Let $\mathbf{Z} = \mathbf{A}^{\frac{1}{2}}\mathbf{Q}$. One can verify $\mathbf{P} = \mathbf{A}^{\frac{1}{2}}\mathbf{Q}\mathbf{Q}^\top\mathbf{A}^{\frac{1}{2}}$. That is to say, the the $k$ largest elements of $\mathrm{diag}(\mathbf{P})$ is equal to the $k$ largest rows of $\mathbf{Z}$ in squred $\ell_2$ norm. Therefore, we can perform thin QR factorization [15] on $\mathbf{A}^{\frac{1}{2}}\mathbf{W}$ to have $\mathbf{Q}$. Then, a simple row $\ell_2$ norm truncation gives $\mathcal{I}$ without explicitly constructing proxy $\mathbf{P}$.

In summary, the orthogonal iteration-like reformulation of Algorithm 2 is presented in Algorithm 3.

---
**Algorithm 3** IPU for general $\mathbf{A}$ (reformulation)
---
1: **procedure** IPU($\mathbf{A}, m, k, d, \mathbf{W}_0$)
2:     compute and cache $\mathbf{A}^{\frac{1}{2}}$;   $t \leftarrow 0$;
3:     **repeat**
4:         $[\mathbf{Q}, \mathbf{R}] \leftarrow$ thin_qr($\mathbf{A}^{\frac{1}{2}}\mathbf{W}_t$);
5:         $\mathbf{Z} \leftarrow \mathbf{A}^{\frac{1}{2}}\mathbf{Q}$;
6:         $\mathcal{I} \leftarrow$ ind. of the $k$ largest rows of $\mathbf{Z}$ in $\ell_2$ norm;
7:         $\mathbf{S} \leftarrow \mathbb{S}_{d,k}(\mathcal{I})$;
8:         $\mathbf{V} \leftarrow m$ first eigenvectors of $\mathbf{A}_{\mathcal{I},\mathcal{I}}$;
9:         $\mathbf{W}_{t+1} \leftarrow \mathbf{S}\mathbf{V}$;   $t \leftarrow t + 1$;
10:    **until** $\mathbf{W}_t = \mathbf{W}_{t-1}$
11:    **return** $\mathbf{W}_t$;
12: **end procedure**
---

To make the paper self-contained and to compare the reformulated Algorithm 3 with the vanilla row-wise truncated orthogonal iteration, we include the latter used in SOAP [43] below:

---
**Algorithm 4** Vanilla row-wise truncated orthogonal iteration [43]
---
1: **procedure** SOAP($\mathbf{A}, m, k, d, \mathbf{W}_0$)
2:     compute and cache $\mathbf{A}^{\frac{1}{2}}$;   $t \leftarrow 0$;
3:     **repeat**
4:         $[\mathbf{Q}, \mathbf{R}] \leftarrow$ thin_qr($\mathbf{A}\mathbf{W}_t$);
5:         $\mathcal{I} \leftarrow$ ind. of the $k$ largest rows of $\mathbf{Q}$ in $\ell_2$ norm;
6:         $\mathbf{S} \leftarrow \mathbb{S}_{d,k}(\mathcal{I})$;
7:         $[\mathbf{V}, \mathbf{R}] \leftarrow$ thin_qr($\mathbf{A}_{\mathcal{I},\mathcal{I}}$);
8:         $\mathbf{W}_{t+1} \leftarrow \mathbf{S}\mathbf{V}$;   $t \leftarrow t + 1$;
9:     **until** $\mathbf{W}_t = \mathbf{W}_{t-1}$
10:    **return** $\mathbf{W}_t$;
11: **end procedure**
---

**Remark I.1.** *It is interesting to see Algorithm 3 has somehow similarity with the vanilla row-wise truncated orthogonal iteration (see [43]). However, we note that these two methods are different significantly in following two aspects: (1) The motivation of IPU is to make full use of the global optimality observation in Algorithm 1, that is, the iterative procedure is specially designed for the FSPCA problem. For truncated orthogonal iteration, the main iterative procedure is basically the well-known orthogonal iteration equipped with a row-wise truncation to project the variables into the feasible domain. (2) The iterative produce of IPU is an ascent algorithm while the truncated orthogonal iteration is not. Please see Section 5.2 and Section 6.2 for theoretical and empirical discussion. It will be interesting to ask whether the proposed algorithm performs better than the vanilla row-wise truncated orthogonal iteration (used in [43] with name SOAP). We conduct extensive experiments on both synthetic and real-world datasets in Section 6.*