[Reviews · NeurIPS 2020]

Review 1

Summary and Contributions: A method for feature-sparse PCA is proposed. The focus is on the optimization aspects, thus reducing the problem to a low-rank approximation (with row-wise sparsity constraints) of a given (deterministic) matrix.

Strengths: According to the simulations, the new method performs very well compared to other competing methods that solve the same optimization problem. The optimization methods looks very simple and clever, and the analysis rigorous, but I admit I don't know the literature well enough to be certain about their novelty.

Weaknesses: The paper is mainly focussed on the optimization aspects. While this is certainly a justifiable subproblem (when only comparing methods that are statistically equivalent), I find that a full evaluation of these methods should look at the statistical aspects as well. I do acknowledge something like this is done in Table 2E,F, although I have trouble understanding the details there. Perhaps the idea is that the statistical properties are inherited from an existing framework using the same objective; if so, please specify.

Correctness: I think so, although I'm not really able to judge. The theory is unfortunately outside of my expertise.

Clarity: I think so.

Relation to Prior Work: The discussion could be improved. For example, a table in the SM comparing with the existing methods would be useful.

Reproducibility: Yes

Additional Feedback:


Review 2

Summary and Contributions: This paper presents a couple of deterministic algorithms for FSPCA problem -- the first algorithm solves FSPCA for the low-rank covariance matrices and the second one is based on an iterative framework and deals with covariance matrices having higher ranks. The authors also provide the correctness of the proposed algorithms, both in terms of accuracy and convergence.

Strengths: I think the paper is well-written and well presented, and I enjoyed reading it. I briefly went over the proofs; they looked correct and were carried out systematically. To the best of my understanding, the idea of using a proxy covariance matrix P in Algorithm 2 is the key to the solid theoretical foundation of this paper. Moreover, the proposed algorithms are deterministic in nature and the corresponding analyses doesn't assume any restrictive assumption on the distribution of the data. Another important point is that the proposed iterative algorithm (Algorithm 2) guarantees strict monotonicity in terms of the objective function value as the number of iterations increase. Finally, the two algorithms also perform exceptionally well as compared to other methods.

Weaknesses: Overall, I think the authors addressed most of the comments/concerns from previous submission.

Correctness: The theoretical claims as well as the experiments look correct.

Clarity: I think the paper is well-written.

Relation to Prior Work: The authors know the background literature well and discussed them thoroughly.

Reproducibility: Yes

Additional Feedback: ======================== After the rebuttal ======================== I have read the authors feedback. ==============================================================


Review 3

Summary and Contributions: The authors propose a new algorithm to solve the sparse constrained PCA problem, where they show that it can achieve global optimality in the low rank case. For the high rank case, they propose an iterative algorithm. Moreover, they demonstrate the convergence and iteration complexity analyses for proposed algorithms. Numerical experiments are provided to show the efficiency of the proposed algorithms.

Strengths: The cardinality constrained sparse PCA is an interesting yet challenging problem. The proposed algorithm provides an alternative approach to the estimation of sparse PCA, with guarantees at different levels of the rank ranges regarding the dimension of the data matrix. The one pass procedure in the low rank case is a strong result since it can converge to the global optimality. In the medium to high rank cases, the proposed iterative algorithm can only guarantee approximate convergence, which is less attractive.

Weaknesses: I have several concerns regarding the results and presentation. For the analysis of medium to high rank cases, the convergence guarantee for IPU seems pretty weak. For example, if the condition number of the data matrix A is small, i.e., the gap between the largest and smallest eigenvalues of A is small, and m is a moderate value compared to r, then the \epsilon can be large (close to 1) so that the estimation error lower bound is large. However, it seems from the experiment that the estimation error is not bad in this case. This may indicate that the analysis of the theory is far from tight and significant improvement is desired. In the high rank case, the theory only guarantee the convergence to a fixed point, but in this cardinality constrained nonconvex problem, a fix point can be far from a true solution based on existing result on this type of factor models. So such an analysis maybe meaningless to discuss the convergence. Another concern is that some further discussion with existing convex/nonconvex sparse PCA algorithms is desired. In particular, it will help the reader understand the pros and cons of the optimization formulation in this paper by comparing their convergence guarantees, computational cost, estimation errors, etc.

Correctness: I did not check all details in the proofs, but it seems correct. The numerical evaluation seems fair and convincing enough.

Clarity: The paper is well presented and clearly written.

Relation to Prior Work: The discussion of prior works lacks some more in-depth comparison with the proposed one, as mentioned in the weaknesses.

Reproducibility: Yes

Additional Feedback: Thanks for the authors' feedback. It clarifies my question on the condition number in Theorem 5.1, and also it justifies the convergence guarantee in Theorem 5.3. I changed my score accordingly.


Review 4

Summary and Contributions: This paper proposed fast provable algorithms for the challenging row sparse PCA problem. The opt objective is basically the fundamental row sparse eigenvalue decomposition problem. Thus, the proposed techniques might be interesting for a lot and could have extensions on a wide range of other machine learning problems, e.g., various spectral algorithms. They run thorough experiments and the new algorithms outperform existing approaches. 1. a simple algorithm (alg. 1) that solves the low-rank FSPCA problem globally 2. a monotonically increasing algorithm (alg. 2) to solve the general case 3. rigorous theoretical results on approximation and convergence 4. very strong and extensive experiments

Strengths: The main contributions are novel and original, to my knowledge. The global optimality of alg.1 is very interesting and somehow surprising. Given the known hardness results of the FSPCA problem, the theorem (4.1) characterizes a subclass of problems that could be perfectly solvable. For high-rank setting, they provide a new iterative minorization-maximization procedure alg.2 by solving a low-rank covariance with alg.1. I found the MM construction here novel as existing results mostly use power method type procedure as the main algorithmic framework. They also provide a clear discussion, though in the appendix, on the differences between theirs and the truncated power method scheme, which is good. The approximation bound and convergence results are nontrivial and sound. On the one hand, I think the authors should highlight the data dependent (on the covariance spectrum) nature of their approximation bound. On the other hand, it is totally okay and useful as well to have a data dependent bound as it is hopeless (SSE-hard, [R1]) to have any constant approximation algorithm for the sparse PCA problem as far as I know. But it is necessary to let the reader know that. The empirical verifications of these bounds on real data are also convincing and informative. Actually, I personally suggest moving these figures 3/4/5 into the main text by putting some settings in Table 2 into the appendix. They also compute some specific examples (exponential/Zipf's decay) with the new bound. The code and sufficient experiment details are provided. It should be easy to reproduce the claimed results. I have reviewed the code and find they aren't the most efficient implementation (the authors have made comments on that in the code). I expect the authors also share the fast version of their algorithms if accepted. The empirical results are strong and extensive. Besides, I found for many real data, alg.1 has already been good enough, though in some cases alg.2 indeed brings additional benefit. So, as alg.1 should run very fast, it is the practicer's choice to trade off the extra computational efforts for the additional improvement from alg.2. REF: --- [R1] Chan, Siu On, Dimitris Papailliopoulos, and Aviad Rubinstein. "On the approximability of sparse pca." Conference on Learning Theory. 2016.

Weaknesses: 1. The author should highlight the data dependent (on the covariance spectrum) nature of their approximation results (maybe in the abstract), though it is totally fine to have a data dependent bound. 2. The reader might be especially interested in the convergence, computing time, and approx ratio on real data. However, these materials are put into the appendix. I personally suggest moving these figures 3/4/5, at least partial of them, into the main paper by instead putting some settings in Table 2 into the appendix. Typos: * line 461: Theorem 5.2 -> Theorem 4.1 * line 644: Problem (A.2) -> Problem 3.1

Correctness: I have carefully checked the proofs of the main theorems, especially these mentioned by prior reviewer/meta-reviewer, and think the proofs are correct.

Clarity: The paper is well written and in good style. Some typos should be fixed.

Relation to Prior Work: Satisfying. The discussions on the algorithmic difference with prior work are clear and good.

Reproducibility: Yes

Additional Feedback:

[Author Response · NeurIPS 2020]

| | | | | | | | | | |
|---|---|---|---|---|---|---|---|---|---|
| SOAP | 0.51 (0.17) | 0.19 (0.09) | 0.00 | 0.46 (0.18) | 0.21 (0.10) | 0.00 | 0.62 (0.15) | 0.14 (0.07) | 0.00 |
| SRT | 0.00 (0.00) | 0.01 (0.02) | 0.35 | 0.00 (0.00) | 0.01 (0.01) | 0.45 | 0.00 (0.00) | 0.00 (0.01) | 0.65 |
| CSSP | 0.49 (0.18) | 0.83 (0.06) | 0.00 | 0.46 (0.16) | 0.84 (0.06) | 0.00 | 0.51 (0.16) | 0.83 (0.07) | 0.00 |
| Go | **0.99 (0.04)** | **0.00 (0.00)** | **0.90** | **0.99 (0.04)** | **0.00 (0.00)** | **1.00** | **0.97 (0.06)** | **0.00 (0.00)** | **0.90** |
| IPU | 0.91 (0.11) | 0.01 (0.01) | 0.50 | **0.99 (0.04)** | **0.00 (0.00)** | **1.00** | **0.97 (0.06)** | **0.00 (0.00)** | **0.90** |

The authors thank the reviewers for their careful readings and insightful and constructive comments. We will improve
the manuscript in the revised version. Below please find our responses to the major points raised.

**To R1:** Thanks for your valuable comments. **On experiments**, with the spike model following [45] with $\mathbf{x}_i =$
$\mathbf{V}\mathbf{z}_i + \sigma\mathbf{w}_i$, where $\mathbf{V} \in \mathbb{R}^{d \times m}$, $\sigma = 0.3$, and Gaussian $\mathbf{z}_i \in \mathbb{R}^m$, $\mathbf{w}_i \in \mathbb{R}^d$. The partial (due to space limit) results are
reported above. More will be included. **On statistical properties,** it could be an easy corollary from Prop. 1.1 of [54].

**To R2:** Thanks for appreciating the contributions of our work!

**To R3:** We appreciate the very detailed and thoughtful comments from the reviewer. We'd like to do some clarifications
here. We proved two types of theoretical results. One is on approximation which controls the absolute error of the
output objective value, that is, accuracy. The other is on convergence which ensures the convergence, which is exact,
and finite time termination of IPU and allows for bounding the approximation error of IPU with Theorem 5.1.

**On tightness when $\kappa = \lambda_1 \lambda_d^{-1} \to 1$.** Thanks! We prove following improved Theorem 5.1, in which $\varepsilon \to 0$ when
$\kappa \to 1$, or $k \to d$, or $\mathbf{A} \to \mathbf{A}_m$. Besides, there is no polynomial algorithm that has small $\varepsilon$ for all $\mathbf{A}$ [8].

**Claim R.1.** *Let the condition number of $\mathbf{A}$ be $\kappa = \lambda_1 \lambda_d^{-1} \geq 1$. In Theorem 5.1 and following corollaries, it holds*

$$\varepsilon \leq \min \left\{ \frac{dG_1}{k}, \frac{dG_2}{m}, 1 - \kappa^{-1}, 1 - \frac{k}{d} \right\}.$$

*Proof.* Using the Poincaré separation theorem in Lemma D.1, we have $\varepsilon \leq 1 - \kappa^{-1}$ by $\mathrm{Tr}(\mathbf{W}_m^\top \mathbf{A} \mathbf{W}_m) \geq$
$\sum_{i=d-m+1}^{d} \lambda_i \geq m \cdot \lambda_d$, and $\mathrm{Tr}(\mathbf{W}_*^\top \mathbf{A} \mathbf{W}_*) \leq \sum_{i=1}^{m} \lambda_i \leq m \cdot \lambda_1 = m \cdot \kappa \lambda_d$. Meanwhile, $\varepsilon \leq 1 - kd^{-1}$ holds by
using $\mathrm{Tr}(\mathbf{W}_m^\top \mathbf{A} \mathbf{W}_m) \geq \mathrm{Tr}(\mathbf{W}_m^\top \mathbf{A}_m \mathbf{W}_m) \geq \frac{k}{d}\mathrm{Tr}(\mathbf{A}_m) \geq \frac{k}{d}\sum_{i=1}^{m} \lambda_i$, and $\mathrm{Tr}(\mathbf{W}_*^\top \mathbf{A} \mathbf{W}_*) \leq \sum_{i=1}^{m} \lambda_i$. $\square$

**On converging to fixed-point.**   In the continuous non-convex optimization literature, it is very common to show
the algorithm converges to a stationary/critical point [52] as the general non-convex optimization are NP-hard even
for computing a local minimizer [50]. However, a stationary point might still be a local maximum/minimum, or
saddle point and far from the global one. Indeed, to our knowledge, it is very difficult (if not impossible) to show
any global convergence guarantee to global optima in general non-convex setting, unless the interested problem has
very special properties, e.g., benign landscape, robust bistability. Moreover, we emphasize that our problem is not a
convex one, and has no known good properties. We are trying to *maximize* a convex objective function with non-convex
combinatorial constraints. So both the objective and the feasible domain bring difficulty. Actually, for our problem,
assuming SSE-hard and NP$\neq$P, it is impossible [8] to have any polynomial running time algorithm that provably returns
$\mathbf{W}$ such that $c \cdot \mathrm{Tr}(\mathbf{W}^\top \mathbf{A} \mathbf{W}) \geq \mathrm{Tr}(\mathbf{W}_*^\top \mathbf{A} \mathbf{W}_*)$ for arbitrary large but finite $c > 1$ and general $\mathbf{A}$. Thus, we think
it is reasonable to have a fixed-point convergence result, especially when the approximation error (accuracy) of the
fixed-point is controlled by Corollary 5.3. Besides, a local analysis near the optima is possible but meaningless, to us,
as it is still NP-hard to ensure the initialization to be in the basin of attraction.

**On related work.**   To the best knowledge of us, there is no directly comparable work in the literature. The most
related work to ours are the very new [54, 53]. [53] proposed algorithm for FSPCA problem with computational
complexity exponential in $m$ and rank($\mathbf{A}$). [53] clearly said (page 4) that their algorithm is of theoretical nature and
may not be practically implementable. In [54], they proposed a heuristic algorithm and a relaxed problem whose
optimal value is upper bound by $(1 + \sqrt{m})^2$ times the FSPCA optimal value. However, [54] provided no rounding
procedure for extracting feasible solution from their relaxation and no approximation guarantee for their heuristic
algorithm. In contrast, our approximation guaranteed algorithm runs in polynomial time and highly implementable
in practice. Besides, [51] proposed an algorithm that runs exponential in the rank($\mathbf{A}$) and $m$ for the *disjoint*-FSPCA
problem that requires the support of different eigenvectors to be *disjoint*, which is clearly different from our setting.
Finally, we note that there are many work [41, 45, 26, 5, 19, 27, 15, 43] that prove results assuming statistical models.
They are not directly comparable to ours as our results are model-free, i.e., same as [54, 53], and applicable for *any*
model and might have applications in other machine learning problems as noted by R4.

**To R4:** Thanks for your insightful and positive comments! We will highlight that in the revised paper and move the
suggested parts to the main paper. The typos have been fixed.

[50] Some np-complete problems in quadratic and nonlinear programming. *Mathematical programming*, 1987.
[51] Sparse pca via bipartite matchings. *NeurIPS*, 2015.
[52] Non-convex optimization for machine learning. *arXiv preprint arXiv:1712.07897*, 2017.
[53] Sparse pca on fixed-rank matrices. *submitted manuscript*, 2019.
[54] Upper bounds for model-free row-sparse principal component analysis. *ICML*, 2020.


[Meta-Review · NeurIPS 2020]

The reviewers all very much liked this paper. The few complaints regarding the condition number and Theorem 5.1 and the convergence analysis to a stationary point in Theorem 5.3, were adequately addressed in the rebuttal. The problem is fundamental, and it is surprising that the authors can demonstrate convergence and iteration complexity for this subclass of problems. The experiments are also very strong. Overall, solid accept.